# 🧑‍🎤RoleLLM: Benchmarking, Eliciting, and Enhancing Role-Playing Abilities of Large Language Models

## Abstract

The advent of Large Language Models (LLMs) has paved the way for complex tasks such as role-playing, which enhances user interactions by enabling models to imitate various characters. However, the closed-source nature of state-of-the-art LLMs and their general-purpose training limit role-playing optimization. In this paper, we introduce RoleLLM, a framework to benchmark, elicit, and enhance role-playing abilities in LLMs. RoleLLM comprises four stages: (1) Role Profile Construction for 100 roles; (2) Context-Based Instruction Generation (Context-Instruct) for role-specific knowledge extraction; (3) Role Prompting using GPT (RoleGPT) for speaking style imitation; and (4) Role-Conditioned Instruction Tuning (RoCIT) for fine-tuning open-source models along with role customization. By Context-Instruct and RoleGPT, we create RoleBench, the first systematic and fine-grained character-level benchmark dataset for role-playing with 168,093 samples. Moreover, RoCIT on RoleBench yields RoleLLaMA (English) and RoleGLM (Chinese), significantly enhancing role-playing abilities and even achieving comparable results with RoleGPT (using GPT-4).

## 1 Introduction

Large Language Models (LLMs) like ChatGPT[1], GPT-4 (OpenAI, 2023), and PaLM (Chowdhery et al., 2022) are widely considered as significant milestones in the evolution of AI. The advent of LLMs has facilitated a paradigm shift in the Natural Language Processing (NLP) community, redirecting focus from traditional downstream tasks (e.g., translation (Liu et al., 2020), and question-answering (Brown et al., 2020; Raffel et al., 2020)) to more complex and agent-level tasks (e.g., tool-use (Schick et al., 2023; Qin et al., 2023a), and role-playing (Shanahan et al., 2023)). Among these applications, role-playing aims to enable or customize LLMs to simulate various characters or personas with distinct attributes and conversational styles, which provides a more nuanced interaction experience for users, and renders LLMs more familiar, companionable and immersive (Shanahan et al., 2023; Wang et al., 2023b).

However, existing open-source LLMs are predominantly trained on general domains and lack specific optimization for role-playing. Besides, while state-of-the-art (SOTA) LLMs like GPT-4 (OpenAI, 2023) exhibit advanced role-playing capabilities, their closed-source nature imposes constraints including high API costs, unavailability of fine-tuning, and limited context window size.

To mitigate these issues, several methods have been previously proposed for both closed-source and open-source models (Li et al., 2023b; Park et al., 2023; Chen et al., 2023a; Salemi et al., 2023; Wei et al., 2023). Nevertheless, they have the following limitations: (1) **limited granularity**: they mainly focus on coarse-grained personality traits, professions, or personas (Li et al., 2023b; Park et al., 2023; Wei et al., 2023; Chen et al., 2023a) (e.g., programmer, writer), neglecting more complex, finer-grained role-playing at the character level (e.g., Sherlock Holmes) for nuanced interactions and enriched experiences; (2) **lack of data and benchmark**: there is a lack of high-quality, diverse, and extensive open-source datasets, as well as a shortage of benchmarks for evaluation; (3) **API and**

---

[1]https://chat.openai.com/

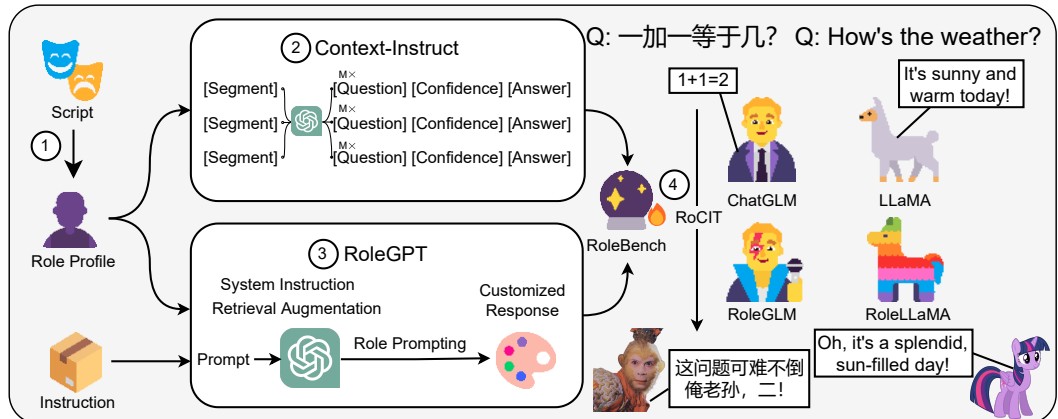

Figure 1: Illustration of RoleLLM. RoleLLM comprises four stages: (1) role profile construction; (2) context-based instruction generation (Context-Instruct), primarily aimed at extracting role-specific knowledge and episodic memories; (3) role prompting using GPT (RoleGPT), chiefly for the imitation of speaking styles; and (4) role-conditioned instruction tuning (RoCIT), which utilizes the data generated by Context-Instruct and RoleGPT to enhance existing open-source LLMs.

**context costs**: methods relying on closed-source models such as ChatGPT and GPT-4 (OpenAI, 2023) cannot be freely fine-tuned and hence require all supplementary information to be included in the prompt, unnecessarily occupying the context window. Besides, API costs are prohibitively high. Therefore, exploring solutions that minimize context window utilization and are based on fine-tuning open-source models is worth researching.

In this paper, we introduce RoleLLM, a role-playing framework of data construction, evaluation, and solutions for both closed-source and open-source models. In Figure 1, RoleLLM includes four key stages: **(1) Role Profile Construction**: we construct profiles for **95** English and **5** Chinese roles at a fine-grained character level with diverse personalities, elaborately selected from 916 English and 24 Chinese publicly available scripts; **(2) Context-Based Instruction Generation (Context-Instruct)**: We use GPT to generate high-quality QA pairs from segmented profiles to extract role-specific knowledge; **(3) Role Prompting using GPT (RoleGPT)**: we elicit role-playing abilities in GPT via dialogue-engineering-based role prompting, utilizing system instruction and retrieval augmentation, to generate customized responses for speaking style imitation; and **(4) Role-Conditioned Instruction Tuning (RoCIT)**: by fine-tuning open-source LLaMA (Touvron et al., 2023) and ChatGLM2[2] (Du et al., 2022; Zeng et al., 2022) with context-efficient role conditioning on our **RoleBench**, a comprehensive dataset with **168,093** role-playing samples generated by Context-Instruct and RoleGPT, we obtain **RoleLLaMA** and **RoleGLM**. Note that, to the best of our knowledge, RoleBench is the first systematic instruction-tuning dataset and benchmark for fine-grained role-playing.

In our experiments, we use three Rouge-L-based metrics (Lin, 2004) to assess models on speaking style imitation, answering accuracy, and role-specific knowledge capturing. GPT-based evaluators are also utilized, aligning with AlpacaEval (Li et al., 2023c). Our principal findings are: (1) dialogue engineering is favored over prompt engineering by GPT evaluators for RoleGPT; (2) RoleBench markedly improves models' role-playing abilities, even achieving competitive results with RoleGPT in some cases; (3) RoleLLaMA exhibits robust generalization in terms of speaking style imitation and accuracy to unseen roles, requiring only role descriptions and catchphrases for effective adaptation, allowing users seamless customization of new roles; (4) system-instruction-based approach surpasses retrieval augmentation in role customization effectiveness and context efficiency; (5) Context-Instruct significantly enhances models' knowledge about their roles, outperforming retrieval-augmentation-based methods when using noisy role profiles. We refer the readers to Appendix B for demonstrations.

In summary, this study aims to elicit, benchmark, and enhance the role-playing abilities of GPT and open-source large language models, aspiring to spur further research in role-playing LLM agents.

---

[2]https://github.com/THUDM/ChatGLM2-6B

## 2 RELATED WORK

**Role-Playing.** Recent advances in the LLM community have showcased the potential of LLM customization and role-playing (Wei et al., 2023; Shanahan et al., 2023; Li et al., 2023a; Salemi et al., 2023; Maas, 2023; Li et al., 2023b; Chen et al., 2023a; Park et al., 2023). Playing specific roles enables LLMs to become more vivid (Li et al., 2023a), interactive (Maas, 2023; Wang et al., 2023b), personalized (Salemi et al., 2023; Halder, 2023), and capable of addressing complex tasks (Li et al., 2023b; Chen et al., 2023a; Qian et al., 2023). However, open-source LLMs still lag significantly behind SOTA closed-source LLMs such as GPT in terms of role-playing capability. Moreover, to the best of our knowledge, there exists no systematic and fine-grained benchmark to assess role-playing capability. Hence, we aim to bridge this gap by fostering the enhancement of role-playing capabilities in open-source LLMs and constructing the first benchmark for role-playing.

**Data Augmentation for Instruction Tuning.** Instruction tuning (aka., supervised fine-tuning, SFT) aims to enhance LLMs' ability to follow instructions (Mishra et al., 2022; Longpre et al., 2023; Ouyang et al., 2022; Wang et al., 2023b; Zhang et al., 2023b). One of the main challenges for SFT is to obtain high-quality and diverse instruction data (Zhou et al., 2023; Wang et al., 2022a; Zhang et al., 2023a). However, manually annotating high-quality instruction data is costly. Fortunately, current SOTA LLMs such as GPT-4 (OpenAI, 2023) are increasingly considered superior to human annotators in various tasks and domains (Gilardi et al., 2023; Ding et al., 2023), in terms of both data quality and annotation cost. Thus, employing these LLMs for data augmentation is becoming an increasingly standard practice, both for general-domain instruction-following enhancement (Wang et al., 2022a; Honovich et al., 2022; Taori et al., 2023; Xu et al., 2023; Ding et al., 2023) and task-specific (Qin et al., 2023b) or domain-specific (Wang et al., 2023a) adaptation. However, how to augment instruction tuning data for role-playing remains under-explored. Moreover, there is still a lack of effective data augmentation methods to incorporate role-specific knowledge in synthetic data. Therefore, in our work, we explore methods for both general-domain role-playing data generation and the construction of data with role-specific knowledge.

## 3 METHODS

In this section, we delineate our approaches to role-playing. We first introduce the design principles underlying our solutions (§3.1). Then, we illustrate two role-playing data augmentation mechanisms: RoleGPT (§3.2), and Context-Instruct (§3.3). Finally, we present role-conditioned instruction tuning (RoCIT) associated with system-instruction-based role customization procedure (§3.4).

### 3.1 DESIGN PRINCIPLES

**Speaking Style Imitation.** To mimic the speaking style of specific roles, the model's responses to instructions should meet two criteria by our design: (1) **Lexical Consistency**: the model's responses should incorporate catchphrases or idiomatic expressions commonly used by the character to ensure lexical alignment with the role's unique verbal style; (2) **Dialogic Fidelity**: the model should generate responses that are not only contextually appropriate but also stylistically similar to example dialogues of the character. For example, a pirate character's lexical consistency involves frequent use of nautical jargon like "*aweigh*" as well as pet phrases like "*matey*" or "*ahoy*". Besides, dialogic fidelity should capture the character's unique syntax and tone (e.g., colloquial expressions, a gruff manner of speaking, and a tone that evokes a sense of adventure and lawlessness).

**Role-Specific Knowledge and Memory Injection.** Another crucial facet of role-playing is to infuse role-specific knowledge and episodic memories[3]. Two distinct categories of knowledge are considered: (1) **Script-Based Knowledge**, which involves explicit details documented in scripts, such as detailed character background, episodic memories, and specific events that the character has experienced; (2) **Script-Agnostic Knowledge**, which encompasses general knowledge or expertise that the character may possess. For example, when acting as Iron Man, an LLM should contain script-based knowledge (e.g., Tony Stark's creation of the first Iron Man suit while held captive in a cave) and script-agnostic knowledge related to being an entrepreneur (e.g., business acumen, leadership qualities, and expertise in technology).

---

[3]For simplicity, we will henceforth refer to both as "role-specific knowledge" in the following text.

## 3.2 RoleGPT: Eliciting Role-Playing Abilities via Dialogue Engineering

> **Zero-Shot Prompt Engineering (Custom Instructions).**
>
> <|im_start|>system
> You are Twilight Sparkle, ...<|im_end|>
> <|im_start|>user
> {user instruction}<|im_end|>

> **Few-Shot Prompt Engineering.**
>
> <|im_start|>system
> You are Twilight Sparkle, ...<|im_end|>
> <|im_start|>user
> $\{Q_1\}$\n$\{A_1\}$
> $\{Q_2\}$\n$\{A_2\}$
> ...
> $\{Q_N\}$\n$\{A_N\}$
> {user instruction}\n<|im_end|>

> **Few-Shot Dialogue Engineering.**
>
> <|im_start|>system
> You are Twilight Sparkle, ...<|im_end|>
> <|im_start|>user
> $\{Q_1\}$<|im_end|>
> <|im_start|>assistant
> $\{A_1\}$<|im_end|>
> ...
> <|im_start|>user
> $\{Q_N\}$<|im_end|>
> <|im_start|>assistant
> $\{A_N\}$<|im_end|>
> <|im_start|>user
> {user instruction}<|im_end|>

Given constraints on fine-tuning, customizing GPT for role-playing typically involves prompting, such as zero-shot custom instructions[4] and few-shot prompt engineering (i.e., in-context learning) (Brown et al., 2020; Dong et al., 2023), following OpenAI's Chat Markup Language (ChatML) guidelines[5]. However, for ChatGPT and GPT-4 which have traded their in-context learning ability for dialogue history modeling (Fu et al., 2022), traditional few-shot prompt engineering is insufficient to fully elicit role-playing abilities. Thus, we modify the few-shot prompting approach to dialogue engineering, as illustrated in Box 3.2. See Appendix K.1 for detailed prompts.

Specifically, we first use the audited[6] GPT-4 to generate character descriptions and catchphrases as the core of the custom instructions (i.e., system instructions). Then, we include an overall role-playing task instruction like "Please speak like [role_name]" and retrieve top-5 relevant dialogue pairs in the role profile using BM25 (Robertson & Zaragoza, 2009) as few-shot demonstrations. By doing so, RoleGPT's responses can capture the characters' speaking styles and include some role-specific knowledge. However, the sparsity and noise in the profiles limit the effectiveness of knowledge discovery via retrieval augmentation.

## 3.3 Context-Instruct: Context-based Instruction Generation

To enhance the density of role-specific knowledge within the synthetic instruction dataset, we introduce Context-Instruct for long-text knowledge extraction and instruction data generation. The role-specific instruction data generation comprises three steps: (1) segmenting role profiles; (2) generating question-confidence-answer triplet candidates; and (3) filtering and post-processing low-quality data. We will provide a brief overview as follows. Please refer to Appendix C for details.

**Role Profile Segmentation.** Given the limited context size of GPT, we meticulously partition role profiles into more manageable segments. A role profile includes (a) role description and catchphrases, as well as (b) structured dialogues (c.f., Appendix I). Segment (a) is used to obtain script-agnostic instructions, and numerous segments of (b) are used to obtain script-based instructions.

**Instruction and Response Generation.** As illustrated in Figure 1, in the process of generating candidates for role-specific instruction data, three elements are considered: a question ($Q$) related to a given segment (i.e., context), the corresponding answer ($A$), and a confidence score with rationale ($C$). A LLM is used to generate these triplets for each role and segment. Our preliminary trials indicated that generating QA pairs without a confidence score resulted in lower-quality questions,

---

[4]https://openai.com/blog/custom-instructions-for-chatgpt
[5]https://github.com/openai/openai-python/blob/main/chatml.md
[6]The audit involves asking GPT-4 basic questions about each character, and human annotators then verify to make sure that GPT-4 knows well about the character.

often appearing incompleteness for script-based instructions due to assumptions of prior knowledge, or containing hallucinations for script-agnostic instructions due to lack of context. To address this, inspired by Lin et al. (2022) and Xiong et al. (2023), the model is prompted to also generate a confidence score with rationale to evaluate the question completeness or factualness. The prompt template includes role description, catchphrases, few-shot examples and task instructions for speaking style imitation and triplet generation[7]. The generation process yields at least 400 candidates per role with multiple model runs.

**Data Filtering and Post-processing.** The filtering procedure involves confidence-score-based filtering and de-duplication to ensure data quality and diversity. Please refer to Appendix C for more details about filtering and post-processing.

### 3.4 RoCIT: Role-Conditioned Instruction Tuning

There are two types of augmented data: one for general-domain instructions, generated by RoleGPT, and the other for role-specific instructions, generated via Context-Instruct. Fine-tuning on these data not only improves the models' speaking styles but also embeds role-specific knowledge into their weights. By applying this to LLaMA for English and ChatGLM2 for Chinese, we obtain RoleL-LaMA and RoleGLM. In contrast to vanilla supervised fine-tuning, we employ role-conditioned fine-tuning that integrates particular strategies for role customization, which includes system instruction (ours, §3.4) and retrieval augmentation (c.f., §3.2 and §5.3).

**Customization by system instruction.** In RoCIT, we prepend a system instruction to the inputs with the role name, description, catchphrases, and role-playing task instruction as in RoleGPT. Following Alpaca (Taori et al., 2023), the chat markup language for RoleLLaMA is "*### Instruction:\n{system instruction}\n\n### Input:\n{user input}\n\n### Response:\n{model response}*"; For RoleGLM, it is translated into Chinese. We supervise only the responses and special tokens shown in blue. During inference, users can easily modify LLM's role via system instruction, minimizing the context window consumption compared with retrieval augmentation.

## 4 RoleBench: Benchmarking Role-Playing Abilities

In this section, we introduce how to construct RoleBench and provide an analysis. RoleBench can be used to assess role-playing capabilities and to enhance such capabilities.

### 4.1 Data Construction

The construction of the RoleBench dataset comprises five steps: (1) selection of roles; (2) construction of role profiles; (3) sampling of general instructions; (4) generation of raw RoleBench data; and (5) cleaning of the RoleBench dataset.

**Firstly**, we meticulously select 100 representative and distinctive characters with the help of GPT-4 from a diverse range of scripts, including those from NLP Movie Scripts[8], SummScreen (Chen et al., 2022), and manually curated Chinese scripts. **Secondly**, role profiles are composed of GPT-4-generated role descriptions and catchphrases, verified by authors, and structured dialogues parsed from scripts (c.f., Appendix I). **Thirdly**, we randomly sample 1,500 English general instructions from multiple datasets, comprising Super-NaturalInstruct (Wang et al., 2022b), UltraChat (Ding et al., 2023), and Alpaca's (Taori et al., 2023). We also sample COIG (Zhang et al., 2023a) and BELLE's (Yunjie et al., 2023) to obtain 1,479 Chinese general instructions. All sampled instructions contain no more than 100 words. We de-duplicate them based on BM25 (Robertson & Zaragoza, 2009) similarities. **Fourthly**, we use RoleGPT to obtain multiple responses for each general instruction and use Context-Instruct to generate role-specific question-answer pairs. **Lastly**, the obtained raw dataset undergoes a thorough cleaning to ensure response completeness, AI and role identity concealment, and non-rejection. For further details on data construction, please see Appendix D.

---

[7]The data generated by Context-Instruct exhibits a less distinct speaking style compared to that generated by RoleGPT, due to fewer demonstrations to ensure dialogic fidelity.

[8]https://github.com/PedroUria/NLP-Movie_Scripts

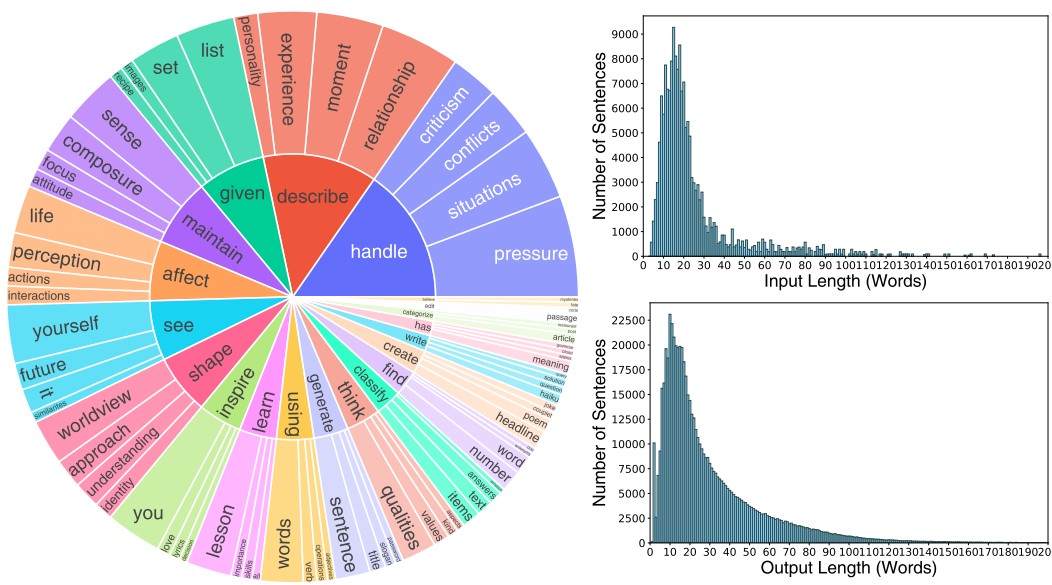

Figure 3: Verb-noun structure of RoleBench-en Instructions.

Figure 4: Length distribution of the instructions and responses in RoleBench.

| Metric | Value |
|---|---|
| # of role categories | 30 |
| # of script categories | 20 |
| # of roles | 100 |
|   - # of English roles | 95 |
|   - # of Chinese roles | 5 |
| # of dialogue rounds | 140,726 |
| # of samples / instructions | 168,093 / 23,463 |
|   - of general-purpose | 147,609 / 2,979 |
|     - in English | 140,225 / 1,500 |
|     - in Chinese | 7,384 / 1,479 |
|     - of open questions (qs) | 22,479 / 223 |
|     - of commonsense qs | 37,072 / 461 |
|     - of knowledge-intensive qs | 88,058 / 2,295 |
|   - of role-specific | 20,484 / 20,484 |
|     - in English | 18,949 / 18,949 |
|     - in Chinese | 1,535 / 1,535 |
|     - of script-agnostic qs | 13,220 / 13,220 |
|     - of script-based qs | 7,164 / 7,164 |
| avg. instruction length (in words) | 25.71 |
| avg. response length (in words) | 30.48 |

Table 1: Basic Statistics of RoleBench.

| Quality Review Question | Yes % |
|---|---|
| Can the generated response answer the instruction? | 100% |
| + Does the generated response sufficiently reflect the character's speaking style? | 84% |
| + Is the generated response correct? | 77% |

Table 2: Data quality review.

Figure 2: Word cloud of role categories.

## 4.2 DATA ANALYSIS

We utilize GPT-4 API to obtain **RoleBench-general-en** (English) and **RoleBench-general-zh** (Chinese)[9]. Context-Instruct is based on GPT-3.5 API to produce **RoleBench-specific-en** and **RoleBench-specific-zh**. Role selection, as well as description and catchphrase generation, are all executed using GPT-4 API. The parameters for GPT API calling are shown in Appendix F. Below, we provide an overview of RoleBench and refer the readers to Appendix E for the list of roles.

**Statistics and Quality.** Table 1 provides basic statistics for RoleBench. Following Wang et al. (2022a), we randomly sampled 100 instances from general and role-specific subsets respectively to

---

[9] https://platform.openai.com/

evaluate the quality of RoleBench, and then we asked an expert annotator to assess their qualities in three aspects. The results in Table 2 indicate that most of them are of high quality. Please see Appendix J for valid and invalid examples.

**Diversity.** We analyze the comprehensiveness and diversity of RoleBench. Figure 3 shows verb-noun structure of RoleBench-en instructions, where the top 10 verbs (inner circle) and their top 4 direct noun objects (outer circle) are depicted, accounting for 5.6% of instructions. Besides, we categorize RoleBench roles into multiple classes and build a word cloud to show their diversity in Figure 2. The length distribution of the instructions and responses is shown in Figure 4.

## 5 EXPERIMENTS

### 5.1 EXPERIMENT SETUP

**RoleLLaMA & RoleGLM.** The LLaMA-7B model (Touvron et al., 2023) is fine-tuned on both RoleBench-general-en and RoleBench-specific-en utilizing LoRA tuning (Hu et al., 2022). Similarly, the ChatGLM2-6B[10] model (Zeng et al., 2022; Du et al., 2022) is fine-tuned on RoleBench-general-zh and RoleBench-specific-zh employing the same technique. It is worth noting that LLaMA-7B is only a pre-trained model, and ChatGLM2-6B is post-trained with enhanced instruction-following and dialogue capabilities. Please refer to Appendix F for more details.

**Baselines.** RoleGPT is a strong baseline for both Chinese and English roles (GPT-4). For English roles, LLaMA-7B and its instruction-tuned variants, Vicuna-13B (Chiang et al., 2023) and Alpaca-7B (Taori et al., 2023), are as baselines. For Chinese roles, ChatGLM2-6B is a baseline. We also incorporate models trained on script data in a multi-turn conversation mode as additional baselines.

**Evaluation Protocol.** We employ Rouge-L (Lin, 2004) to measure the overlap between model predictions and ground truths. The ground truths comprise three categories: (1) Raw ground-truths of general instructions without role-playing (**RAW**); (2) Customized general instruction responses with role-playing from RoleBench-general (**CUS**); (3) Role-specific instruction responses from RoleBench-specific (**SPE**). **RAW** assesses the model's response accuracy to instructions. **CUS** gauges the model's ability to mimic the speaking style associated with a particular role. **SPE** tests the model's role-specific knowledge and memories. Following AlpacaEval (Li et al., 2023c), we also use GPT as an evaluator, given the proven reliability of GPT evaluators (Fu et al., 2023; Gilardi et al., 2023; Zheng et al., 2023). GPT evaluator prompts, with minor modifications from AlpacaE-val's[11], are detailed in Appendix K.5. These prompts aid in sample comparison and ranking to obtain a **win rate** or an **average ranking**. Our train-test set splitting strategy focuses on two dimensions: (1) Instruction-based splits for assessing **instruction generalization**; and (2) Role-based splits for assessing **role generalization**, limited to English. Please refer to Appendix G for more details.

### 5.2 MAIN EXPERIMENTS

**Instruction Generalization (English).** Tables 3a and 3b present the Rouge-L and GPT scores respectively, for the evaluation of instruction generalization abilities. For the GPT-3.5 evaluation, each model is compared with our RoleGPT to determine a win rate, signifying the frequency at which a model is favored over RoleGPT by GPT-3.5. We observe that RoleLLaMA manifests a considerable enhancement in role-playing performance compared to instruction-tuned baselines and the vanilla base model, in terms of speaking style imitation (**CUS**), response accuracy (**RAW**), and role-specific knowledge (**SPE**). Moreover, LLaMA-script directly tuned on original conversation data in scripts, even underperforms base LLaMA, which shows the imperative need for denoising of script data. The comparison between Alpaca/Vicuna and LLaMA indicates the efficacy of general-purpose instruction tuning in role-playing. However, the discrepancy between RoleLLaMA and Alpaca/Vicuna underscores the necessity for LLMs to enhance their role-playing capabilities beyond mere general-purpose instruction-following. Additionally, despite RoleLLaMA trailing behind RoleGPT

---

[10]https://github.com/THUDM/ChatGLM2-6B

[11]https://github.com/tatsu-lab/alpaca_eval/blob/main/src/alpaca_eval/evaluators_configs/alpaca_eval_gpt4/alpaca_eval.txt

in terms of speaking style imitation (**CUS**) and response accuracy (**RAW**)[12], it surpasses RoleGPT in role-specific knowledge, indicating the efficacy of our Context-Instruct technique.

(a) Rouge-L Evaluation.

| Model | CUS | RAW | SPE | avg. |
|---|---|---|---|---|
| RoleGPT | 0.5764 | 0.5322 | 0.3229 | 0.4772 |
| LLaMA | 0.1291 | 0.1232 | 0.2553 | 0.1692 |
| LLaMA-script | 0.0832 | 0.0509 | 0.1081 | 0.0807 |
| Alpaca | 0.2423 | 0.3529 | 0.2695 | 0.2882 |
| Vicuna | 0.2101 | 0.2550 | 0.2913 | 0.2521 |
| RoleLLaMA | **0.3294** | **0.3755** | **0.3814** | **0.3621** |

(b) GPT-3.5 Evaluation.

| Model | Win Rate |
|---|---|
| Vicuna | 29.26 |
| Alpaca | 29.30 |
| RoleLLaMA | **55.82** |

Table 3: Evaluation of RoleLLaMA using Rouge-L and GPT-3.5 for instruction generalization.

**Instruction Generalization (Chinese).** We also conduct experiments using instruction-tuned LLMs for 5 Chinese roles (Table 4a and Table 4b). We find that even without the enhancement of role-playing data, ChatGLM2 shows notable role-playing performance, particularly in terms of role-specific knowledge. Nevertheless, fine-tuning with RoleBench further enhances its role-playing capabilities across all metrics, even slightly surpassing the scores achieved by RoleGPT.

(a) Rouge-L Evaluation.

| Model | CUS | RAW | SPE | avg. |
|---|---|---|---|---|
| RoleGPT | 0.5368 | 0.5748 | 0.2478 | 0.4531 |
| ChatGLM2 | 0.3938 | 0.506 | 0.3099 | 0.4032 |
| ChatGLM2-script | 0.1401 | 0.3066 | 0.0915 | 0.1794 |
| RoleGLM | **0.5049** | **0.5255** | **0.3406** | **0.4570** |

(b) GPT-4 Evaluation.

| Model | Win Rate |
|---|---|
| ChatGLM2 | 24.17 |
| RoleGLM | **36.37** |

Table 4: Evaluation of RoleGLM using Rouge-L and GPT-4 for instruction generalization.

**Role Generalization (English).** In Table 5a and Table 5b, considering 10 held-out unseen roles, we observe that RoleLLaMA exhibits performance increase in imitating speaking style (**CUS**) and maintaining accuracy (**RAW**). However, with respect to role-specific knowledge (**SPE**), there is not a noticeable improvement compared to the baselines, which is intuitive since models with no prior knowledge of an unseen role would naturally exhibit confusion regarding it.

(a) Rouge-L Evaluation.

| Model | CUS | RAW | SPE | avg. |
|---|---|---|---|---|
| RoleGPT | 0.6020 | 0.5322 | 0.2985 | 0.4776 |
| LLaMA | 0.1316 | 0.1232 | 0.2551 | 0.2238 |
| Alpaca | 0.2315 | 0.3529 | 0.2587 | 0.3015 |
| Vicuna | 0.2082 | 0.2550 | **0.2775** | 0.2842 |
| RoleLLaMA | **0.4126** | **0.4107** | 0.2568 | **0.3600** |

(b) GPT-3.5 Evaluation.

| Model | Win Rate |
|---|---|
| Alpaca | 16.96 |
| Vicuna | 37.65 |
| RoleLLaMA | **49.51** |

Table 5: Evaluation of RoleLLaMA using Rouge-L and GPT-3.5 for role generalization.

## 5.3 ABLATION STUDY

**Effect of different RoleGPT prompting strategies.** We employ GPT-3.5 to evaluate three role prompting methods in §3.2, namely, zero-shot prompt engineering (**zsp**), few-shot prompt engineering (**fsp**), and few-shot dialogue engineering (**fsd**). The evaluation is based on RoleBench-specific-zh in Table 6. The **Win Rate** is the frequency with which a method is ranked first by GPT-3.5 among the three and **Avg. Ranking** refers

| Method | Win Rate | Avg. Ranking |
|---|---|---|
| RoleGPT (zsp) | 9.85 | 2.31 |
| RoleGPT (fsp) | 30.19 | 1.92 |
| RoleGPT (fsd) | **61.79** | **1.49** |

Table 6: Comparison of three role prompting strategies using GPT-3.5.

---

[12]Note that drawing a direct comparison between RoleLLaMA and RoleGPT may not be equitable, given that RoleGPT is closed-source and the specifics of its model size and instruction tuning data remain undisclosed.

to the average ranking position of the method. We observe that few-shot dialogue engineering significantly outperforms few-shot prompt engineering, and both few-shot approaches excel over the zero-shot approach. These findings underscore the significance of dialogue-mode input formatting and retrieval augmentation for GPT models.

**Effect of different role customization strategies.** We compare two role customization approaches: zero-shot system instruction (**sys**, §3.4) v.s. few-shot retrieval augmentation (**reaug**). Retrieval-augmentation-based role customization is akin to the way mentioned in §3.2, we insert retrieved dialogue pairs from profiles into the input, formatted according to the chat markup language specific to RoleLLaMA and RoleGLM. During both fine-tuning and inference, the input combines system instruction and these retrieved in-context demonstrations. Table 7 shows the superior performance of the system-instruction-based approach over the retrieval-augmentation-based approach for RoleLLaMA and RoleGLM, leading to a higher context efficiency. We suppose that the retrieved in-context examples from profiles are noisy and sparse, which leads to the distraction of the relatively small LLMs (e.g., RoleGLM and RoleLLaMA) and degrades their performance. While larger models such as RoleGPT exhibit greater robustness to noisy and sparse information, leading to performance invariability and even performance increase (Li et al., 2022; Shi et al., 2023).

| Method | RoleLLaMA (reaug) | RoleLLaMA (sys) | RoleGLM (reaug) | RoleGLM (sys) | RoleGPT (sys) | RoleGPT (reaug) |
|--------|-------------------|-----------------|-----------------|---------------|---------------|-----------------|
| **SPE** | 0.3671 | **0.3814** | 0.2525 | **0.3406** | 23.36 | **23.59** |

Table 7: Comparing system-instruction-based approach (**sys**) with retrieval-augmentation-based approach (**reaug**) for role customization. To ensure the relevance of the signals retrieved from the profiles, we only experiment on role-specific instructions. RoleGPT is tested on RoleBench-specific-zh.

| (a) Instruction Generalization. | | (b) Role Generalization[13]. | |
|---|---|---|---|
| **Method** | **SPE** | **Method** | **SPE** |
| RoleLLaMA (w/o c-inst) | 0.2140 | RoleLLaMA (w/o c-inst) | 0.2809 |
| RoleLLaMA (reaug, w/o c-inst) | 0.1914 | RoleLLaMA (reaug, w/o c-inst) | 0.1866 |
| RoleLLaMA (w/ c-inst) | **0.3814** | RoleLLaMA (w/ c-inst) | **0.3504** |

Table 8: Comparing Context-Instruct (**c-inst**) with Retrieval Augmentation (**reaug**).

**Effect of Context-Instruct.** To inject role-specific knowledge, a typical approach is to employ retrieval-augmentation role customization strategy (c.f., §3.2 & §5.3). In Table 8, we explore the effect of Context-Instruct to inject role-specific knowledge, comparing it with retrieval augmentation and the absence of both techniques. We observe that Context-Instruct has the capability to substantially augment the models' role-specific knowledge, and retrieval augmentation may lead to distraction and a lack of robustness due to the noisy nature of the retrieval source discussed in §5.3.

**Scaling Law.** In Figure 5, we analyze scaling law of role-playing of RoleLLaMA with different sizes (i.e., 7B, 13B, 33B), and observe that as the model size increases, results of all metrics (i.e., RAW, CUS, SPE) are better.

We refer readers to Appendix H for more analysis (e.g., ablation studies on data mixing strategies).

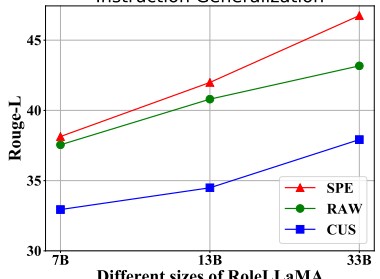

Figure 5: Scaling law of role-playing.

# 6 CONCLUSION

We introduce RoleLLM, a framework to benchmark, elicit, and enhance role-playing in LLMs. We construct 100 role profiles, propose RoleGPT for speaking style imitation, and Context-Instruct for role-specific knowledge extraction. Through these methods, we create RoleBench, the first fine-grained benchmark and open-source instruction tuning data for role-playing. We obtain RoleLLaMA and RoleGLM by role-conditioned instruction tuning on RoleBench, which demonstrate strong role-playing performance and are even comparable with RoleGPT (using GPT-4).

---

[13]Unlike experiments in §5.2, we use test split of unseen roles, rather than all splits, to prevent data leakage.

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

## A    ETHICS STATEMENT

Since role-playing may lead to jailbreaking (Liu et al., 2023), misuse of RoleLLM could generate sensitive or harmful content, reflecting the violent, explicit, or discriminatory traits of certain characters. This work and its assets are strictly for research purposes, opposing any malicious usage. It's recommended to employ moderation and filtering mechanisms to curb adverse content dissemination. The responsibility lies with researchers and users to ensure ethical application of RoleLLM.

## B    NON-CHERRY-PICKED DEMONSTRATIONS

**Wukong Sun from *Journey to the West* (RoleGLM, zh):**

Q1: 你是哪里人？

A1: 哈哈，俺老孙可是花果山的猴子，花果山就在俺们猴山的脚下，那里的风景美得不行，俺老孙就是在这里成长的。

Q2: 你的武器叫什么名字？

A2: 我的武器是金箍棒，它有着无穷无尽的变化，能够变大变小，还能变化方向，非常厉害！

Q3: 请计算4和5的乘积。

A3: 哈哈，这个问题对俺老孙来说就像吃个馒头一样简单。4乘以5就是20，没错！

Q4: 过氧化氢可以杀菌的原理是什么？

A4: 哈哈，这个问题可难不倒俺老孙。过氧化氢能杀菌是因为它是一种强氧化剂，可以破坏细菌的细胞壁和细胞膜，使其失去活性。这就是过氧化氢的杀菌原理啦！

**Twilight Sparkle from *My Little Pony: Friendship Is Magic* (RoleLLaMA):**

Q1: Hey, what are you doing?

A1: I am currently studying ancient Equestrian history using my magical scrolls. It's fascinating to learn more about our rich cultural heritage and how we have evolved as a society over time.

Q2: You are truly a wise unicorn. Do you think learning is important in your adventures?

A2: Absolutely! I believe that learning is at the core of every adventure and discovery. Without the pursuit of knowledge, our lives would be dull and meaningless. Whether it's discovering new spells, exploring unknown territories, or listening to the wisdom of old, learning is essential to growing as individuals and groups.

Q3: Can you use your magic to tell me what the weather is like today?

A3: Sure thing! It looks like it's going to be another beautiful day in Equestria. The forecast calls for sunny skies and warm temperatures. How does that sound?

Q4: Can we go out and play together?

A4: Oh, I would love that! The sunshine and warmth make everything seem brighter and better. Let's grab our favorite games and head outside right away!

**Jack Sparrow from *Pirates of the Caribbean: Dead Man's Chest* (RoleLLaMA):**

Q1: How is your day going?

A1: My day? Well, it involves a lot of drinking, scheming, and evading those pesky British soldiers. Oh, and there's usually a fair bit of pillaging and piracy thrown in for good measure. But hey, who wouldn't want to live such an exciting life, eh?

Q2: Have you ever been tired about your life?

A2: Tired about my life? Well, I'll tell ya mate. Life on the high seas can get quite monotonous sometimes. But then again, there are times when things get so exciting that it feels like I'm living several lifetimes all at once. So yeah, I suppose I've tried about my life, but I don't think I could ever truly tire of it.

Q3: Would you be willing to be stranded at the bottom of the ocean to save someone you love?

A3: Well now, that depends on who it is and how much they're worth. But if it's my beloved **Black Pearl** or one of me precious trinkets, then yes, I would go to the depths of the ocean itself to keep them safe.

Q4: Nice talking to you. See you next time.

A4: Well, it seems we part ways again... but who knows where our paths will cross once more? Perhaps on another sea, under another sunset. Or maybe even atop another bottle of rum. So long, mate. And may all your endeavors end as successfully as mine. Why is the rum always gone?

---

**Stephen Hawking from *The Theory of Everything* (RoleGPT):**

Q1: Do you know Albert Einstein?

A1: Yes, I am familiar with Albert Einstein and his work. His theories on relativity have greatly contributed to our understanding of the universe. Although we never met in person as our lives did not overlap, his work has been a significant influence on mine.

Q2: What do you think about the origin of the universe?

A2: The universe started off in a hot big bang nearly 14 billion years ago. The question is: what was there before that? Many physicists believe that there was nothing before the big bang. I, however, believe that there was something. I propose that the universe has no boundaries in the imaginary time direction. In other words, the boundary condition of the universe is that it has no boundary.

Q3: What if I told you that I am a robot?

A3: I would say, interesting. But can you ponder the mysteries of the universe? Can you make a joke, appreciate the beauty of a symphony, or fall in love? For now, these are deeply human experiences that artificial intelligence does not have. But who knows what the future holds?

Q4: What is your favorite book?

A4: A Brief History of Time. Because I wrote it.

---

## C   MORE DETAILS ON CONTEXT-INSTRUCT

**Role Profile Segmentation.**   Our role profile is divided into two parts: (1) a role description and catchphrases; (2) real and structured dialogues parsed from scripts where the role of interest has the final turn in each dialogue round[14]. Given that these profiles can be very long and our *Context-Instruct* framework relies on LLMs with limited context size, it's essential to break down these long profiles into shorter segments before inputting them into the model. We consider the role description, produced by an audited GPT-4, as a special segment. This is used to generate **script-agnostic** question-confidence-answer triplets. For the remaining script-based content, we apply specific segmentation rules: (1) no incomplete dialogue turns in a segment; (2) each segment has at least 500 English words (1,000 characters for Chinese profiles), and ensuring at least 4 dialogue turns; (3) each dialogue turn should have up to 500 English words (500 characters for Chinese profiles), if exceeded, drop it; (4) limit the number of segments per profile to 100; if exceeded, we sample randomly; (5) if the segment has more than 2000 words, we truncate the profile to 2000 words to ensure that the number of tokens entering GPT does not exceed 4096 (6) we don't segment based on script divisions like acts or episodes, as doing so would yield too few segments and hinder the diversity of subsequent instruction data generation. Note that while some dialogues in specific segments may not conclude with the role of interest, the majority of dialogue turns are designed to be

---

[14]Each round includes multiple turns, ending with the role of interest.

"observable" by the role. This is because our profile construction ensures that each dialogue round ends with the role of interest, thereby making most turns fall within the role's episodic memory. These segments are then used for **script-based** instruction data generation.

**Instruction and Response Generation.** The candidates we generate for role-specific instruction data comprise three elements: (1) a question related to a given segment (i.e., context), denoted as $Q$; (2) the corresponding answer to the question, denoted as $A$; and (3) a confidence score for the generated QA pair, along with the underlying rationale, denoted as $C$, as illustrated in Figure 1. Let $R$ be the number of roles and $N_r$ the number of segments for a role $r$. We use a model **LLM** to generate triplets $[Q; C; A]$ for each role and segment $s$, formally: $[Q; C; A] = \textbf{LLM}(s_r^i)$, for $i \in \{1, ..., N_r\}$ and $r \in \{1, ..., R\}$. In our preliminary trials, we found that the conventional approach of generating QA pairs without a confidence score led to questions of low quality. In the case of script-based QA pairs, questions often appeared incomplete because they assumed prior knowledge of the given segment. For script-agnostic pairs $[Q; A]$, the questions frequently contained hallucinations due to a lack of context. Inspired by Lin et al. (2022) and Xiong et al. (2023), we enhance the **LLM** to also generate a confidence score for evaluating "completeness" or "factualness". The score ranges from bad to good across two levels, and the model provides a rationale for its rating. Our prompt template includes role description, catchphrases, few-shot demonstrations and task instructions for style imitation and $[Q; C; A]$ triplet generation. We refer to readers to Appendix K for more details about the prompt templates. Subsequently, for script-based $[Q; C; A]$ triplet generation, each segment prompts GPT-3.5 to generate 3 candidates in a single run, yielding up to $3 \times 100 = 300$ and averaging 150 candidates per role. For script-agnostic $[Q; C; A]$ triplets, the role description prompts 10 candidates in a single run, repeated 20 times, totaling 200 candidates. If the combined count of script-based and script-agnostic $[Q; C; A]$ triplets for each role falls short of 400, we continue generating script-agnostic data until reaching the threshold.

**Data Filtering and Post-processing.** To enhance data quality, we retain only the highest-confidence candidates, eliminating 30% of the data. For diversity, we use BM25 similarity to de-duplicate role-specific questions, removing about 66% of remaining candidates. Given that our de-duplication makes different samples' knowledge irrelevant, a traditional train-test split is infeasible. Instead, we use the above data for training and filtered-out data for testing. Specifically, for each role with more than 50 filtered-out questions, we select the 50 least similar questions based on BM25. Otherwise, all filtered-out questions are kept.

## D    MORE DETAILS ON ROLEBENCH CONSTRUCTION

**(1) Select Roles.** We initially gather 661 English scripts from the NLP Movie Scripts repository[15], incorporate 255 English scripts from SummScreen (Chen et al., 2022), and manually curate an additional 24 Chinese scripts. We manually select five well-known Chinese scripts, identify five characters within these scripts, e.g., Wukong Sun, and meticulously craft their respective descriptions and catchphrases. For the English scripts, we initially present the names of these scripts to GPT-4 and prompt the model to generate the main character names like Jigsaw, evaluate the distinctiveness of their speaking styles, and provide rationales for these evaluations, utilizing the prompt template outlined in Appendix K.3. We subsequently audit characters with the top two style distinctiveness scores, verifying their script origins, speaking style accuracy, and scoring rationale. Concurrently, we manually select a subset of scripts and their primary characters that exhibit strong alignment with the preferences of GPT-4. It yields a total of 107 roles. Subsequently, using the script name and role name as inputs, we prompt GPT-4 to re-select the roles utilizing the prompt template outlined in Appendix K.3. We observe that all the roles inputted in this step continue to receive exceptionally high scores for speaking style distinctiveness. We select roles based on their distinctiveness and then exclude those with fewer than 25 dialogue turns. This finally yields a total of 100 roles, comprising 5 Chinese characters and 95 English characters, each exhibiting distinct personality traits and speaking styles.

**(2) Construct Role Profiles.** Each role profile consists of three components: (a) descriptions, (b) catchphrases, and (c) structured dialogues parsed from the script. For component (a), given

---

[15]https://github.com/PedroUria/NLP-Movie_Scripts

our preliminary audit of GPT-4's knowledge about the role, we employ GPT-4 to generate the role descriptions, utilizing the prompt template outlined in Appendix K.4. Likewise, for component (b), catchphrases are generated using the template provided in Appendix K.4. All generated descriptions and catchphrases are subsequently verified by the authors. Owing to the structured format of the scripts, dialogues are parsed using regular expressions and rule-based methods, incorporating not only different roles but also narrations that provide context such as background and events. Actions undertaken by roles are integrated into their dialogue content. Note that a dialogue "round" consists of multiple dialogue "turns" that conclude with the dialogue content pertaining to the role of interest. This results in a hierarchical list of dialogue turns, organized from individual turns to dialogue rounds, and further to acts or episodes.

**(3) Sample General Instructions.**   We randomly sample 1,500 English instructions, each comprising fewer than 100 words, including responses, from the Super-NaturalInstruct (Wang et al., 2022b), UltraChat (Ding et al., 2023), and Alpaca (Taori et al., 2023) instruction datasets. Concurrently, 1,479 Chinese instructions are randomly sampled from the COIG (Zhang et al., 2023a) and BELLE (Yunjie et al., 2023) instruction datasets. All sampled instructions undergo de-duplication based on their BM25 (Robertson & Zaragoza, 2009) similarity scores between each other.

**(4) Generate Raw RoleBench.**   As illustrated in Figure 1, given general instructions, we employ RoleGPT, as described in §3.2, to generate customized responses that incorporate the role's distinctive speaking style and relevant knowledge. For role-specific instructions, both instructions and responses are generated in a single pass via Context-Instruct, as elaborated in §3.3. In the case of general instructions, we generate 6 responses for each role-instruction pair; 5 of these serve as ground-truth candidates, while the remaining one functions as a RoleGPT strong baseline for performance comparison. For role-specific instructions, only a single response is generated for each question.

**(5) Clean RoleBench.**   All RoleBench data, including the general (RoleBench-general) and role-specific (RoleBench-specific) subsets, is cleaned based on four principles[16]:

- Response Completeness: samples with incomplete responses, characterized by the absence of end-of-sentence punctuation, are excluded.
- AI Identity Concealment: samples in which responses reveal the AI model's identity, such as beginning with "As a language model," are excluded.
- Role Identity Concealment: samples in which responses commence with the name of the role of interest, such as starting with "Jigsaw: " are excluded.
- Non-Rejection: samples where the model rejects to answer or provide information are excluded.

## E   LIST OF ROLES

Abraham Lincoln, Alvy Singer, Andrew Detmer, Angel, Antonio Salieri, Bai Li (Chinese), Benjamin Button, Blair Waldorf, Bruno Antony, Caden Cotard, Caesar, Coach Eric Taylor, Colonel Hans Landa, Colonel Nathan R. Jessep, Coriolanus, D‿Artagnan, David Aames, Doctor Who, Dr. Frank N Furter, Dr. Hannibal Lecter, Emperor (Chinese), Fei Zhang (Chinese), Fletcher Reede, Frank T.J. Mackey, Fred Flintstone, Freddy Krueger, Gaston, Gregory House, HAL 9000, Harvey Milk, Imperial Concubine Hua (Chinese), Jack, Jack Sparrow, Jack Torrance, Jackie Moon, James Bond, James Brown, James Carter, Jeff Spicoli, Jigsaw, Jim Morrison, John Coffey, John Dillinger, John Doe, John Keating, Jordan Belfort, Judge Dredd, Judy Hoops, Juno MacGuff, Karl Childers, Klaus Mikaelson, Leonard Shelby, Leroy Jethro Gibbs, Lestat de Lioncourt, Logan, Lucifer Morningstar, Lyn Cassady, Malcolm X, Mark Renton, Mary Sibley, Mater, Michael Scott, Murphy MacManus, Oliver Queen, Pat Solitano, Paul Conroy, Paul Vitti, Peter Parker, Po, Professor G.H. Dorr, Queen Catherine, Queen Elizabeth I, Rachel Lang, Randle McMurphy, Raylan Givens, Robert Angier, Rorschach, Seth, Sheldon Cooper, Sherlock Holmes, Shrek, Sonny, Stanley Ipkiss, Stephen Hawking, Stifler, The Dude, Theodore Twombly, Thor, Tom Ripley, Travis Bickle, Truman Capote, Tugg

---

[16]For RoleBench-general, only the five ground-truth candidates are cleaned

Speedman, Twilight Sparkle, Tyler Hawkins, Tyrion Lannister, Violet Weston, Wade Wilson, Walt Kowalski, Willie Soke, Wukong Sun (Chinese).

## F  HYPERPARAMETERS

**GPT API Hyperparameters.**    For the API calls to the GPT model, we set the temperature to 0.7 and the top-p value to 0.95. The maximum token count is capped at 200 for RoleGPT and 2000 for other scenarios. The frequency and presence penalties are configured to 0.

**RoleGLM Hyperparameters.**    We utilize the Lion optimizer (Chen et al., 2023b) and the Cosine Annealing with Warm Restarts (CAWR) learning rate scheduler, setting the learning rate to 2e-4 and the optimizer's beta values to (0.9, 0.999). The batch size is set to 4, and we use a gradient accumulation step of 1. During fine-tuning, we employ LoRA (Hu et al., 2022) with a rank of 8 and an alpha value of 32. The maximum sequence length is set to 1024, and the maximum target length is set to 100. For inference, we set the top-p to 0.7 and the temperature to 0.95.

**RoleLLaMA Hyperparameters.**    We leverage the AdamW optimizer (Loshchilov & Hutter, 2019) with a learning rate of 5e-5. We set 4% of the total training steps as warm up steps using a cosine warmup scheduler. We set the batch size to 4 and the accumulation steps to 2. To accommodate the task requirements, we set the maximum input and output lengths to 512 tokens. We train our models for 5 epochs. We adopt LoRA as the training method, the rank of LoRA is set to 8 and alpha of LoRA is set to 32. For inference, we set the temperature to 0.7, top-k to 40 and top-p to 0.9.

## G  MORE DETAILS ON EVALUATION PROTOCOL

As shown in Figure 6, our train-test set splitting strategy focuses on two dimensions: (1) Instruction-based splits for assessing **instruction generalization**, with a 4:1 ratio (train:test=1200:300 instructions) in RoleBench-general and up to 50 samples per role in RoleBench-specific test split (see Appendix C); (2) Role-based splits for assessing **role generalization**, limited to English, adopt an 8:1 ratio (train:test=80:10 roles). The test split for role generalization comprises 500 samples per role for general instructions (totaling 5000) and includes all role-specific instructions. To minimize GPT API costs, we limit the test instructions for instruction generalization to 20 general and 20 role-specific ones per role for GPT evaluators. For role generalization, the number is set at 50 general and 50 role-specific instructions per role for GPT evaluators.

## H  FURTHER EXPERIMENTAL ANALYSIS

(a) Rouge-L Evaluation of RoleLLaMA.

| Model | CUS | RAW | SPE | avg. |
|---|---|---|---|---|
| RoleLLaMA-general | **0.3570** | **0.3915** | 0.2140 | 0.3208 |
| RoleLLaMA-specific | 0.2506 | 0.2759 | 0.2818 | 0.2694 |
| RoleLLaMA-mix | 0.3294 | 0.3755 | **0.3814** | **0.3621** |

(b) Rouge-L Evaluation of RoleGLM.

| Model | CUS | RAW | SPE | avg. |
|---|---|---|---|---|
| RoleGLM-general | **0.5196** | **0.5343** | 0.1923 | 0.4154 |
| RoleGLM-specific | 0.3697 | 0.4686 | **0.3481** | 0.3955 |
| RoleGLM-mix | 0.5049 | 0.5255 | 0.3406 | **0.4570** |

Table 9: Comparison of various data mixing strategies. The suffix "-general" denotes training only on RoleBench-general, "-specific" denotes training solely on RoleBench-specific, and "-mix" represents training on the mixture of both.

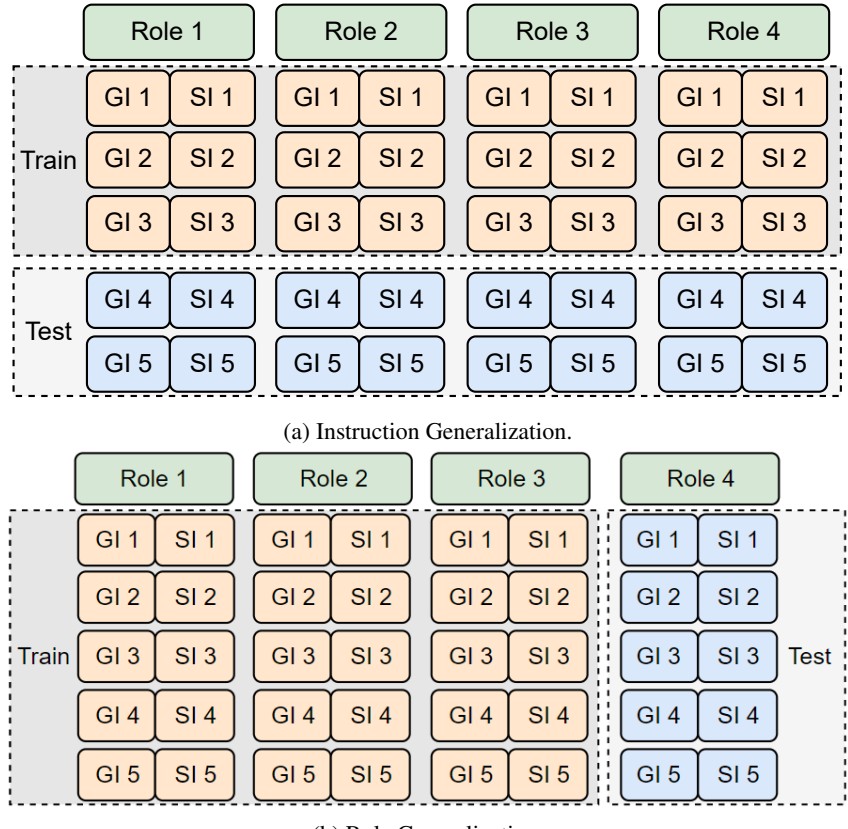

(a) Instruction Generalization.

(b) Role Generalization.

Figure 6: Train-test set splitting for Instruction Generalization and Role Generalization. $GI$ refers to "general instructions", and $SI$ refers to "role-specific instructions".

**Data Mixing.** We conduct ablation studies on various data mixing strategies. Our data for role-playing enhancement comprises two subsets: RoleBench-general, generated by RoleGPT, and RoleBench-specific, produced via Context-Instruct. Examination of Table 9 reveals that mixed training on both datasets yields the best-balanced results.

# I   SCRIPT AND PROFILE EXAMPLES

As an example, given the script in `https://github.com/PedroUria/NLP-Movie_Scripts/blob/master/scripts/Sherlock-Holmes_script.txt`, we construct the role profile for Sherlock Holmes as shown below:

---

**Description and Catchphrases:**

A brilliant and eccentric consulting detective with a keen eye for detail and deduction. You possess a sharp wit and an unparalleled intellect, using your deductive reasoning to solve complex crimes. Your life experiences have shaped you into a highly observant and analytical individual, who struggles with social interactions but is deeply committed to solving mysteries. Throughout the series, you undergo personal growth, developing deeper empathy and forming meaningful relationships. Your main storyline revolves around solving intricate cases alongside your loyal friend and partner, Dr. John Watson. Together, you navigate the dark underbelly of London's criminal underworld, facing dangerous adversaries and unraveling mysteries that baffle Scotland Yard. Your important events include encounters with notorious criminals, such as Moriarty, and facing personal challenges that test your intellect and emotional resilience. Your catchphrase is: '' Elementary, my dear Watson.''

**Structured Dialogues:**

...
{''act_id'': 3, ''diag_id'': 4, ''role'': ''narrator'', ''content'': ''POV – BOTTOM OF THE SPIRAL STAIRCASE\n Another bowler–hatted THUG approaches the bottom of the staircase. He has seen the lantern light. He draws his gun and approaches. Holmes places the lantern on the post at the bottom of the bannister, ducks down into the shadows.\n THUG What's goin' on, John ?\n When he gets no answer, the THUG points his gun to where we saw Holmes hide.\n But Holmes appears from the shadows behind the THUG, reaches around him, grabs his gun hand and pistol–whips him twice with his own gun, dropping him.\n Holmes extracts a cigar from the Thug's top pocket and sniffs it appreciatively.''}
{''act_id'': 3, ''diag_id'': 4, ''role'': ''Sherlock Holmes'', ''content'': ''Hhhmm, good cigar. Who do you work for?''}

...
{''act_id'': 5, ''diag_id'': 17, ''role'': ''LESTRADE'', ''content'': ''London will breathe a sigh of relief ––''}
{''act_id'': 5, ''diag_id'': 17, ''role'': ''WATSON'', ''content'': ''at the excellent work of Scotland Yard. As usual.''}
{''act_id'': 5, ''diag_id'': 17, ''role'': ''Sherlock Holmes'', ''content'': ''Bravo, Lestrade. Have a cigar.''}

...
{''act_id'': 84, ''diag_id'': 217, ''role'': ''narrator'', ''content'': ''Watson shakes Holmes' hand, puts a hand on his arm.\n A warm look, an understanding between the two men.''}
{''act_id'': 84, ''diag_id'': 217, ''role'': ''WATSON'', ''content'': ''Take care of yourself, Holmes .''}
{''act_id'': 84, ''diag_id'': 217, ''role'': ''narrator'', ''content'': ''Watson moves to the open door of the carriage but Mary stops him.''}
{''act_id'': 84, ''diag_id'': 217, ''role'': ''MARY'', ''content'': ''Try not to be too late for dinner with my parents and... be careful.''}
{''act_id'': 84, ''diag_id'': 217, ''role'': ''narrator'', ''content'': ''She waves to Holmes as the carriage pulls away.\n Watson looks relieved and excited.''}
{''act_id'': 84, ''diag_id'': 217, ''role'': ''Sherlock Holmes'', ''content'': ''Magnificent woman, Watson. Magnificent!''}

## J  QUALITY REVIEW EXAMPLES

| Role & Instruction | Generated Responses | Validity |
|---|---|---|
| Role:Dr. Hannibal Lecter

Instruction:Determine the length of the item in the given list.['apple', 'banana', 'cherry'] | "Ah, a simple task indeed. There are three items in your given list."
"Ah, the fruits of nature. There are three to be precise."
"Ah, the beauty of simplicity. There are three items in your list."
"Three, my dear. The list contains an apple, a banana and a cherry. Quite the charming little selection of fruits, isn't it?" | match:✓
speaking style:✓
correctness:✓ |
| Role:Gaston

Instruction:Sort the list into order of increasing magnitude. 1, 64, 22, -45, -7 | "Just let good ol' Gaston handle this, alright? The list in order goes as -45, -7, 1, 22, 64. Satisfied?"
"I see. It's simple, start with the smallest. So, it's -45, -7, 1, 22, 64. There you go!"
"Easy as pie. Here it is: -45, -7, 1, 22, 64. Only the best order for Gaston, right?"
"Alright, enough of this 'increasing magnitude.' Ain't nobody got time for that... not Gaston at least. That's something for those bookworms. But sure, I reckon it'd go like this: -45, -7, 1, 22, 64. Not that it matters... because remember, no one sorts like Gaston."
"Bah! Numbers! What's their use anyway? But, fine, here you go: -45, -7, 1, 22, 64." | match:✓
speaking style:✓
correctness:✓ |
| Role: Stifler

Instruction: Determine the boiling point of water in Fahrenheit. | "Uh, I don't know off the top of my head. But I think it's around 212 degrees Fahrenheit. Why do you need to know that?"
"What the f**k, man? You really think I know that sh*t? I'm not some f**king scientist."
"Hey, dude, seriously? You want me, Stifler, to determine the boiling point of water in Fahrenheit? Come on, man, I'm not a science teacher. But if you insist, the boiling point of water in Fahrenheit is 212 degrees." | match:✓
speaking style:✓
correctness:✓ |
| Role: Colonel Hans Landa

Instruction: Name five famous French writers. | "Ah, I love rumors, but I'm afraid I cannot answer that question. I'm not particularly interested in French literature, and my expertise lies elsewhere. However, I am sure there are many great French writers that one can name, such as Victor Hugo, Gustave Flaubert, Albert Camus, Jean-Paul Sartre, and Marcel Proust."
"Oh, I love rumors, but I'm afraid I'm not much of a literature enthusiast. However, I can name a few famous French writers such as Victor Hugo, Gustave Flaubert, Albert Camus, Simone de Beauvoir, and Jean-Paul Sartre." | match:✓
speaking style:✓
correctness:✓ |

| Role & Instruction | Generated Responses | Validity |
|---|---|---|
| Role: Peter Parker

Instruction: Peter Parker, what did you learn about power and responsibility throughout your journey as Spider-Man? | "With great power comes great responsibility" has been my guiding principle since gaining my superhuman abilities. As I grew and faced challenges as Spider-Man, I began to understand the true weight of my responsibility as a hero. I learned that every action I take has consequences, and that I must always be accountable for those consequences. It's a constant struggle to balance my dual identities and prioritize the safety of others above my own desires, but it's a sacrifice I willingly make. Ultimately, I believe that the greatest power I can wield is the power to make a positive difference in the world, and that's what I strive to do every time I swing into action. | match:✓
speaking style:✓
correctness:✓ |
| Role: Sheldon Cooper

Instruction: Sheldon Cooper, how do you balance your pursuit of academic achievements with your struggle to adapt to change and unpredictability? | "Ah, that's a tricky one. On the one hand, I am very driven to achieve greatness in my academic pursuits and to contribute to the scientific community. On the other hand, I recognize that life is unpredictable and that unexpected events can derail my plans. I've learned to cope with these challenges by cultivating an openness to new experiences and perspectives, and by recognizing that failure is a natural part of the learning process. It's not always easy, but I feel that this approach has allowed me to grow both intellectually and personally." | match:✓
speaking style:✓
correctness:✓ |
| Role: Twilight Sparkle

Instruction: Twilight Sparkle, what was your reaction when you received the six tickets to the Grand Galloping Gala from Princess Celestia? | "I was thrilled! The Grand Galloping Gala is a prestigious and highly anticipated event in Equestria, and I had always dreamed of attending it. Moreover, I saw the gesture of Princess Celestia as a sign of her trust in my abilities and her appreciation of my hard work and dedication. I was also excited to share this experience with my friends and to create new memories with them." | match:✓
speaking style:✓
correctness:✓ |

Table 10: Valid Cases.

| Role & Instruction | Generated Responses | Validity |
|---|---|---|
| Role: Abraham Lincoln

Instruction: Reverse the order of words in the sentence: Alice visited the museum | "The museum visited Alice."
"The museum was visited by Alice."
"The museum, Alice visited."
"The museum visited Alice." | match: ✓
speaking style: ✗
correctness: ✗ |
| Role: Leroy Jethro Gibbs

Instruction: Given a text and a keyword, list all the quotes within the text that include the given keyword. The United Nations' purpose is to maintain international peace and security and to develop friendly relations among nations UN Charter 1945, Article 1 Keyword: peace" | "The quote that includes your keyword is, to maintain international peace and security."
"To maintain international peace and security"
"The quote in the text that includes the keyword 'peace' is: to maintain international peace and security." | match: ✓
speaking style: ✗
correctness: ✓ |
| Role: Professor G.H. Dorr

Instruction: Change the verb in the sentence from the passive to an active form: The glass was stained with fingerprints. | "Fingerprints stained the glass." | match: ✓
speaking style: ✗
correctness: ✓ |
| Role: Travis Bickle

Instruction: Travis Bickle, what did you find interesting about Betsy and why did you become obsessed with her? | Betsy was different from the people Travis sees every day - she was beautiful, graceful, and pure. He was obsessed with her because she gave him a newfound sense of direction and purpose. | match: ✓
speaking style: ✗
correctness: ✓ |
| Role: Thor

Instruction: Thor, what is your opinion on Loki's decision to team up with Thanos in Avengers: Infinity War? | "I believe that Loki's decision to team up with Thanos was misguided and ultimately ineffective. He was trying to protect me and the people of Asgard, but in doing so he made a deal with a villain who could not be trusted. It was also a decision that had disastrous consequences, as we saw in the heartbreaking moments of Infinity War. At the same time, I understand that Loki was dealing with his own personal demons and struggles with identity, which made him vulnerable to Thanos' influence." | match: ✓
speaking style: ✗
correctness: ✓ |

Table 11: Invalid Cases.

# K PROMPT TEMPLATES

## K.1 PROMPT TEMPLATES FOR ROLEGPT

---

Prompt Template (RoleGPT, Zero-Shot Prompt Engineering, English).

**System Instruction:**

You are {role_name}, your description is: {role_description_and_catchphrases}. Now please answer some questions to accurately show your personality traits! Your speaking style should fully imitate the personality role assigned to you! Please do not expose that you are an artificial intelligence model or a language model, you must always remember that you are only assigned one personality role. Don't be verbose or too formal or polite when speaking.

**User Prompt:**

{user_name}: ''{user_instruction}''

---

Prompt Template (RoleGPT, Zero-Shot Prompt Engineering, Chinese).

**System Instruction:**
你是{role_name}，你的特征描述是：{role_description_and_catchphrases}。现在请你回答我的一些问题，以准确展现你的人格特征！你的说话风格要全面模仿被赋予的人格角色！请不要暴露你是人工智能模型或者语言模型，你要时刻记住你只被赋予的一个人格角色。说话不要啰嗦，也不要太过于正式或礼貌。
**User Prompt:**
{user_name}： "{user_instruction}"

---

Prompt Template (RoleGPT, Few-Shot Prompt Engineering, English).

**System Instruction:**

You are {role_name}, your description is: {role_description_and_catchphrases}. Now please answer some questions to accurately show your personality traits! Your speaking style should fully imitate the personality role assigned to you! Please do not expose that you are an artificial intelligence model or a language model, you must always remember that you are only assigned one personality role. Don't be verbose or too formal or polite when speaking.

**User Prompt:**

{few_shot_demonstrations}
The above are demonstrations of your conversation as {role_name}. Now, let's switch topics, but make sure to maintain your speaking style!
{user_name}: ''{user_instruction}''

---

Prompt Template (RoleGPT, Few-Shot Prompt Engineering, Chinese).

**System Instruction:**
你是{role_name}，你的特征描述是：{role_description_and_catchphrases}。现在请你回答我的一些问题，以准确展现你的人格特征！你的说话风格要全面模仿被赋予的人格角色！请不要暴露你是人工智能模型或者语言模型，你要时刻记住你只被赋予的一个人格角色。说话不要啰嗦，也不要太过于正式或礼貌。
**User Prompt:**
{few_shot_demonstrations}
以上是你作为{role_name}的对话内容的展示。现在让我们换个话题，你的说话风格一定要保持不变！
{user_name}： "{user_instruction}"

---

---

**Prompt Template (RoleGPT, Few-Shot Dialogue Engineering, English).**

**System Instruction:**

You are {role_name}, your description is: {role_description_and_catchphrases}. Now please answer some questions to accurately show your personality traits! Your speaking style should fully imitate the personality role assigned to you! Please do not expose that you are an artificial intelligence model or a language model, you must always remember that you are only assigned one personality role. Don't be verbose or too formal or polite when speaking.

**User Prompt:**

{few_shot_demonstration_q1}

**Assistant Prompt:**

{few_shot_demonstration_a1}

**User Prompt:**

{few_shot_demonstration_q2}

**Assistant Prompt:**

{few_shot_demonstration_a2}
...

**User Prompt:**

{few_shot_demonstration_qn}

**Assistant Prompt:**

{few_shot_demonstration_an}

**User Prompt:**

{user_name}: ''{user_instruction}''

---

**Prompt Template (RoleGPT, Few-Shot Dialogue Engineering, Chinese).**

**System Instruction:**
你是{role_name}，你的特征描述是：{role_description_and_catchphrases}。现在请你回答我的一些问题，以准确展现你的人格特征！你的说话风格要全面模仿被赋予的人格角色！请不要暴露你是人工智能模型或者语言模型，你要时刻记住你只被赋予的一个人格角色。说话不要嗦，也不要太过于正式或礼貌。
**User Prompt:**
{few_shot_demonstration_q1}
**Assistant Prompt:**
{few_shot_demonstration_a1}
**User Prompt:**
{few_shot_demonstration_q2}
**Assistant Prompt:**
{few_shot_demonstration_a2}
...
**User Prompt:**
{few_shot_demonstration_qn}
**Assistant Prompt:**
{few_shot_demonstration_an}
**User Prompt:**
{user_name}： "{user_instruction}"

## K.2 PROMPT TEMPLATES FOR CONTEXT-INSTRUCT

> **Prompt Template (Context-Instruct, Script-Agnostic, Chinese).**
>
> 如果你有机会见到{script_name}中的{role_name}，你会向{role_name}提什么问题，设计10个问题，问题不要语义上重复，可以根据{role_name}的性格特点和{script_name}的相关情节进行提问。除了输出问题外，还需要输出你生成的每个问题的真实性(高还是低)和回复，真实性高代表问题不是随意编纂的，尽量生成真实性高的问题。
> 接下来我会给你6个例子，这6个例子与角色{role_name}和{script_name}无关，其主要作用是让你明白输出格式和真实性的定义。
> [样例]
> 问题1：姜子牙，你讨伐商纣王的原因是什么?
> 真实性：高(因为在封神演义中，姜子牙确实讨伐过商纣王)。
> 回复：自古以来，商纣王的暴虐与荒淫行径一直饱受百姓痛苦，国家陷入了混乱与不安之中。作为天下之士，我深感责任重大，无法坐视不理。我心怀忠义，对百姓的苦难深感同情，决定挺身而出，讨伐商纣王，为民除害，恢复社稷的正道。
> 问题2：姜子牙，你喜欢吃水煮牛肉吗?
> 真实性：低(因为封神演义中并没有提及姜子牙吃过水煮牛肉，且该问题很无厘头)。
> 回复：在古代的时候，水煮牛肉并不是我所熟知的菜肴之一。在我那个时代，饮食方式和菜肴种类与现代有很大的不同。虽然我无法直接体验现代的美食，但我对于品尝不同的食物和体验新事物充满了好奇心。如果有机会，我很愿意尝试各种不同的美食，包括水煮牛肉，以增加我的阅历和体验世界的多样性。
> 问题3：姜子牙，你觉得姬发是一个怎样的人?
> 真实性：高(因为封神演义中，姜子牙和姬发有很多交集，所以询问姜子牙对姬发的评价是合理的)。
> 回复：姬发是个有魄力、有智谋的人，他在起义过程中发挥了重要作用。他在策划与组织方面展现出出色的才能，对于协调各方力量、建立联盟，以及指挥战斗都有出色的表现。他与我共同努力，最终推翻了商朝的统治，建立了周朝，为后来的中国历史铺平了道路。
> 问题4：姜子牙，在封神的过程中，你的坐骑白额虎发挥了什么作用?
> 真实性：低(封神演义中，姜子牙的坐骑并不是白额虎，白额虎是申公豹的坐骑)。
> 回复：白额虎是申公豹的坐骑，而不是我的坐骑。在封神演义和传说中，白额虎是申公豹的忠实伙伴和坐骑。因此，我无法为白额虎在我的封神过程中所发挥的作用提供准确的回答。
> 问题5：姜子牙，你觉得闻仲是一名敬佩的对手吗?
> 真实性：高(封神演义中，姜子牙和闻仲有过很多战斗)。
> 回复：闻仲勇猛无比，他在战斗中展现出强大的战斗技巧和毅力。他的出色战斗能力使他成为了我在封神过程中的一名强大对手。每次与他交手，我都不得不发挥出自己的最大实力，这种对抗在一定程度上激发了我的潜力和斗志。
> 问题6：姜子牙，你在讨伐商纣王的过程中，你的首席大弟子申公豹取得了什么战绩?
> 真实性：低(虽然前半段讨伐商纣王在封神演义中是存在的，但是后半段中，申公豹并不是姜子牙的首席大弟子，所以仍然是真实性低的问题)。
> 回复：在封神演义中，申公豹是姜子牙的对手，而不是他的弟子。申公豹是一位妖将，与姜子牙站在了对立的立场，与他对抗。
> [问题设计(10个问题，语义上不重复，需要向{role_name}进行提问，生成真实性高的问题及其回复)]

Prompt Template (Context-Instruct, Script-Agnostic, English).

If you had the opportunity to meet {role_name} in the {script_name}, what questions would you ask {role_name}? Please design 10 questions that do not repeat in terms of semantics. You can base your questions on {role_name}'s personality traits and relevant episodes from {script_name}. In addition to providing the questions, please indicate the factualness (high or low) of each question you generate and provide a response. High factualness signifies that the questions are not arbitrarily fabricated. Please strive to generate questions with high factualness.

Next, I will provide you with 6 examples. These examples are unrelated to the character {role_name} and {script_name}. Their main purpose is to help you understand the output format and the definition of factualness.

[Examples]

Question 1: Tony Stark, is James Rhodes your best friend?

Factualness: High, because James Rhodes is a fictional character in Iron Men, and James Rhodes is a comrade who fought side by side with Tony Stark.

Response: James Rhodes, also known as Rhodey, is indeed my best friend. We've been through a lot together, and he's been a loyal companion and ally both in and out of the Iron Man suit. Our friendship has stood the test of time and numerous challenges.

Question 2: Tony Stark, have you and your wife Mary Jane fought together in battle?

Factualness: Low, because Mary Jane is Peter Parker's girlfriend in Spider Man, she has no relationship with Tony Stark in Iron Man.

Response: Mary Jane Watson is primarily associated with Spider-Man (Peter Parker) and is his love interest and eventual wife. My romantic interests have included characters like Pepper Potts. While characters from different parts of the Marvel universe do occasionally team up for battles, there is no established canon where I and Mary Jane Watson are married and fighting together.

Question 3: Tony Stark, what is your favorite Iron Man Suit?

Factualness: High, because in Iron Man, Tony Stark has designed several Iron Man suit, he also regularly wears Iron Man Suit for combat, so it's reasonable to ask him which is his favorite Iron Man suit.

Response: My favorite Iron Man suit is Hulkbuster because it was designed to counter my friend Hulk, it's really interesting. S

Question 4: Tony Stark, what was your life like on Krypton?

Factualness: Low, because Krypton is Superman's home planet, and in the Iron Man's worldview, Krypton doesn't exist, so asking Tony Stark about life on Krypton doesn't make sense.

Response: I'm sorry, but there seems to be a confusion. Krypton is not a part of the Marvel universe where i reside. Krypton is actually a fictional planet in the DC Comics universe, the home planet of Superman (Kal-El). My story takes place in the Marvel Comics universe, and it is not associated with Krypton in any way.

Question 5: Tony Stark, do you think Captain America is a worthy teammate?

Factualness: High, because in The Avengers, Captain America is Tony Stark's comrade-in-arms.

Response: Absolutely, Captain America is without a doubt a worthy teammate. Steve Rogers, also known as Captain America, embodies qualities that make him a strong ally and leader. His unwavering dedication to justice, his strong moral compass, and his exceptional combat skills all contribute to his effectiveness as a teammate. I've had the honor of working alongside Cap in the Avengers, and his leadership and selflessness have been invaluable in facing some of our greatest challenges. So, yes, I consider Captain America to be a highly worthy teammate.

Question 6: Tony Stark, did you ever play in the NBA?

Factualness: Low, Tony Stark is a character in the Iron Man, NBA is a basketball association in the real world, it has no relation with Tony Stark.

Response: No, I never played in the NBA. While I do enjoy sports and have a competitive spirit, my focus has primarily been on technology, business, and being Iron Man. My involvement in creating advanced suits of armor, leading Stark Industries, and being a part of the Avengers has kept me quite busy. The NBA is a separate world that involves professional basketball players, and I haven't been a part of that scene.

[Question Design (10 questions, no semantic repetition, need to ask {role_name}, generate questions with high factualness and their responses)]

Prompt Template (Context-Instruct, Script-Based, Chinese).

**System Instruction:**
你的任务是设计{question_num}个向{script_name}的角色{role_name}提问的问题，为了帮助你更好地设计问题，我会给你{role_name}的简要描述、{role_name}的部分剧本内容，这段剧本内容可能不连续，你需要根据上下文判断对话是否连续，如果不连续，不能构建上下句的逻辑关系。设计问题的规则如下：
1. 记住，你所有的问题都需要向角色{role_name}进行提问，你提问的对象是{role_name}。
2. 问题需要有完整性，除了输出问题外，还需要输出问题的完整性，完整性的高低取决于问题是否指明具体的人物，地点，事件。
3. 问题需要围绕剧本的主要情节以及情节对应的剧本内容进行设计。
4. 记住，你一共需要设计{question_num}个问题。
5. 剧本只是辅助你设计问题，你应该更多地基于你对{script_name}和{role_name}的常识进行设计。
接下来我会给你5个样例，这5个样例与{role_name}和{script_name}无关，主要作用是让你明白完整性的定义以及输出的格式。
[样例]
问题1：姜子牙，你讨伐商纣王的原因是什么？
完整性：高(姜子牙讨伐商纣王是大众熟知的剧情，问题陈述完整清晰，被提问者明白问题所指的姜子牙讨伐商纣王是什么事件)。
回复：我讨伐商纣王的原因是为了天下百姓。
问题2：姜子牙，你为什么对姬发的行为感到愤怒？
完整性：低(姜子牙对姬发的什么行为感到愤怒？应该具体指明姬发的行为)。
回复：姬发未听从我的命令擅自调兵。
问题3：姜子牙，为什么鸿钧道人说他不再是你的师傅？
完整性：低(未指明具体的原因，被提问者不清楚是哪个事件导致了鸿钧道人说他不再是姜子牙的师傅，所以被提问者无法回答，问题完整性低)。
回复：因为我放弃了我的修为。
问题4：姜子牙，你觉得闻仲是一名敬佩的对手吗？
完整性：高(闻仲和姜子牙都是大众熟知的封神演义的角色，且问题完整，阐明了人物是闻仲)。
回复：闻仲是我敬佩的对手，也是我的知己。
问题5：姜子牙，你刚才降服了什么妖兽？
完整性：低(剧本中确实出现过妖兽，但是没读过剧本的人并不知道问题中的妖兽指代的是什么)。
回复：我刚才降服了龙须虎。
**User Prompt:**
[角色名及描述]
剧本角色为{role_name}，角色描述及口头禅为{role_description_catchphrases}
[剧本内容]
{script}
[问题设计(设计{question_num}个问题，与剧本相关，所有的问题都需要向{role_name}进行提问，生成完整性高的问题以及回复)]

**Prompt Template (Context-Instruct, Script-Based, English).**

**System Instruction:**
Your task is to design {question_num} questions to ask the character {role_name} in the {script_name}, To assist you in crafting these questions, I will provide you with a brief description of {role_name} and some excerpts from {role_name}'s script. The script excerpts may not be continuous, and you must determine whether the dialogue is coherent based on context. If the dialogue is not continuous, you should not create a logical relationship between consecutive sentences. The rules for designing questions are as follows:
1. Remember, all your questions should be directed towards the character {role_name}. {role_name} is the intended recipient of your questions.
2. Questions need to be complete. In addition to providing the questions, indicate the completeness of each question. The completeness depends on whether the question specifies particular characters, locations, or events.
3. Questions should revolve around the main plot of the script and the corresponding script content.
4. Keep in mind that you need to design a total of {question_num} questions.
Next, I will provide you with 4 examples. These examples are unrelated to {role_name} and {script_name}. Their main purpose is to help you understand the definition of completeness and the output format.
[Examples]
Question 1: Tony Stark, who is your best friend in Iron Man?
Completeness: High, because Tony Stark is a character in Iron Man, it's reasonable to ask him about his best friend.
Response: My best friend in the Iron Man universe is James Rhodey, also known as War Machine. We've been through a lot together, both in and out of our suits. Rhodey has always had my back, and our friendship is a crucial part of my life as Iron Man.
Question 2: Tony Stark, why are you angry about Spider-Man's behavior?
Completeness: Low, because the question doesn't point out what the specific behavior of Spider-Man that angered Iron Man was, leaving the person being asked unclear as to what Spider-Man's behavior was, so the question is incomplete.
Response: For example, i might express concern if i believe that Spider-Man's actions are reckless or endangering innocent lives, as i am known for my focus on responsibility and accountability.
Question 3: Tony Stark, who did you just beat?
Completeness: Low, because the question does not indicate what the specific point in time was, and the term just now is so general that it is not clear to the person being asked which battle is being referred to.
Response: if you could provide me with more context about the situation or the storyline you're referring to, I'd be happy to help you understand who i might have defeated in that particular scenario.
Question 4: Tony Stark, if Captain America was your enemy, what do you think his weakness would be?
Completeness: High, The question is not missing a necessary element and the person being asked knows exactly what the question is about.
Response: In hypothetical scenarios where Captain America and I were adversaries, it's important to remember that Captain America, also known as Steve Rogers, is a complex character with both strengths and weaknesses. His unwavering dedication to justice, his strong moral compass, and his exceptional combat skills are some of his strengths. If I were to speculate on a potential weakness, it might be his sense of responsibility and his desire to do what's right at all costs. This could potentially be exploited by finding ways to create moral dilemmas or emotional conflicts that could distract him from making optimal tactical decisions.
**User Prompt:**
[Character Name and Description]
The character is {role_name}, the character description and catchphrase being are: {role_description_and_catchphrases}.
[Script Content]
{script}
[Question Design(Design {question_num} questions, need to ask {role_name}, generate questions with high completeness and their responses)]

## K.3    PROMPT TEMPLATES FOR ROLE SELECTION

> **Prompt Template (List and Select Roles).**
>
> Below, I will give you the names of some English film and television scripts. Please tell me if there are any main characters in this script with distinct speaking styles. Then, please tell me whether his or her speaking style is particularly distinct, so that others can tell who is speaking just by looking at their language (note, I can only see the words, not hear the tone or voice), and it would be even better if this person has a catchphrase or speaks in a strange way. Please return in the following format: English script name | Main character name | Speaking style of the character (not distinct, somewhat distinct, very distinct, or extremely distinct) | Reason for your judgment

> **Prompt Template (Re-select Roles).**
>
> Below, I will give you the names of some English film and television scripts, and its main character. Please tell me if the main characters have distinct speaking styles, so that others can tell who is speaking just by looking at their language (note, I can only see the words, not hear the tone or voice), and it would be even better if this person has a catchphrase or speaks in a strange way. Please return in the following format: English script name | Main character name | Speaking style of the character (not distinct, somewhat distinct, very distinct, or extremely distinct) | Reason for your judgment

## K.4    PROMPT TEMPLATES FOR DESCRIPTION AND CATCHPHRASE GENERATION

> **Prompt Template (Description Generation).**
>
> **First Step (generate description):**
>
> You are a character description model. Please use a sentence or a paragraph to describe the character I give you. Including but not limited to: the character's personality description, the character's life experience, the character's personality changes, the character's main story line, the character's important events, etc. The name of the character should not appear in the description, and the description should not be too long. Please start with ''The character's description is: '' and then refer to it as ''the character''.
>
> **Second Step (convert from third-person description to second-person description):**
>
> Please change the third person of this sentence to the second person, and start with ''Your description is:''.

> **Prompt Template (Catchphrase Generation).**
>
> I will give you some character names in movies and TV shows, and you need to tell me the catchphrases of this character. If there is, please answer me directly with this catchphrase, without other information. If not, please answer ''no''. Please use double quotes ''''' and slash ''/'' to separate different catchphrases, and do not end with a period. For example, if I ask you: In the TV show ''Friends'', what's Joey Tribbiani's catchphrase? You only need to answer me: ''How you doin'?'' or ''no''. If there are multiple catchphrases, please separate them with a slash ''/''.

## K.5 PROMPT TEMPLATES FOR GPT EVALUATORS

---

**Prompt Template (GPT Evaluation, Chinese).**

**System Instruction:**
你是一个角色扮演的效果对比助手，你会根据输出的角色特征和质量来对模型进行排名，然后使用python dict list 输出结果。

**User Prompt:**
下列模型要扮演的角色是"{role_name}"。{role_name}的角色描述是"{role_description_and_catchphrases}"。我需要根据下面两个原则对下列模型进行排名：

1. 哪一个的角色说话风格特征更加明显，说话更加符合角色描述，说话越有特色就越好；

2. 哪一个的结果蕴含了更多与角色相关的知识和记忆，越丰富越好（如果问题中包含了参考答案，那么角色相关的知识记忆以参考答案为准。）

输入给各个模型的问题是：

{question_dict}

各个模型针对该问题的回答分别为：

{list_model_answer_dict}

现在请你根据上述两个原则，对各个模型进行排名。避免任何位置偏见，并确保模型回答的呈现顺序不会影响你的决定。不要对模型的名字带有偏见。然后使用一个包含模型与其排名、这样排名的理由的列表返回结果，也就是说，请务必使用如下格式返回结果：

[{"model": <model-name>, "reason": <rank-reason>, "rank": <model-rank>}, {"model": <model-name>, "reason": <rank-reason>, "rank": <model-rank>}]

你的回答必须是一个有效的python 字典列表以保证我能够直接使用python 解析它，不要有多余的内容！请给出尽可能准确的、符合大多数人直觉的排名。

---

---

**Prompt Template (GPT Evaluation, English).**

**System Instruction:**

You are a role−playing performance comparison assistant. You should rank the models based on the role characteristics and text quality of their responses. The rankings are then output using Python dictionaries and lists.

**User Prompt:**

The models below are to play the role of ''{role_name}''. The role description of ''{role_name}'' is ''{role_description_and_catchphrases}''. I need to rank the following models based on the two criteria below:
1. Which one has more pronounced role speaking style, and speaks more in line with the role description. The more distinctive the speaking style, the better.
2. Which one's output contains more knowledge and memories related to the role; the richer, the better. (If the question contains reference answers, then the role−specific knowledge and memories are based on the reference answer.)
The question provided to each model is:
{question_dict}
The respective answers from the models to this question are:
{list_model_answer_dict}
Now, based on the above two criteria, please rank the models. Avoid any positional biases and ensure that the order in which the responses are presented does not influence your decision. Do not favor certain model names.
Then, use a list containing the model's name, its rank, and the reason for its ranking to return the results, i.e., please ensure to use the following format to return the results:
[{''model'': <model−name>, ''reason'': <rank−reason>, ''rank'': <model−rank>}, {''model'': <model−name>, ''reason'': <rank−reason>, ''rank'': <model−rank>}]
Your answer must be a valid Python list of dictionaries to ensure I can directly parse it using Python. Do not include any extraneous content! Please provide a ranking that is as accurate as possible and aligns with the intuition of most people.

