# OpenReview forum: "RoleLLM: Benchmarking, Eliciting, and Enhancing Role-Playing Abilities of Large Language Models"
_ICLR.cc/2024/Conference — ICLR 2024 Conference Withdrawn Submission_

### Official Review · Reviewer_nGiR · 2023-11-01

**Soundness:** 3 good
**Presentation:** 3 good
**Contribution:** 3 good
**Rating:** 6
**Confidence:** 3

**Summary:**

This paper proposes a framework called RoleLLM to benchmark, elicit, and enhance the role-playing abilities LLM. It proposes a method to elicit role-playing abilities in closed-source LLMs such as GPT-4 using dialogue engineering and custom instructions. Also, this paper introduces the first benchmark and open-source dataset for role-playing.

**Strengths:**

This paper designs four stages for role playing: role profile construction, context-based instruction generation, role prompting using GPT, and role-conditioned instruction tuning
This paper proposes RoleBench, the first fine-grained benchmark and open-source dataset for role-playing.

**Weaknesses:**

This article has some issues with insufficient comprehensiveness in its evaluation. The performance of GPT-3.5 may be limited to evaluate these metrics given that GPT-4 is now commonly used for evaluating the results of weaker models. Lack of human evaluation to compare these role generation results.

**Questions:**

Why is RoleLLaMA's performance so much lower than RoleGPT? Could the author please provide some analysis?

**Details Of Ethics Concerns:**

To learn certain characters may require the appropriate authorization.

---

> ### Author Response · Authors · 2023-11-18
> **Response - 1**
>
> Q1: The performance of GPT-3.5 may be limited to evaluate these metrics given that GPT-4 is now commonly used for evaluating the results of weaker models.
>
> A1: Thanks for your comments!
>
> After utilizing GPT-4 to assess the RoleGLM model before the deadline for submission, we depleted our entire quota for the GPT-4 API, leaving us with only GPT-3.5 as an option. However, we recently acquired additional GPT-4 API access and have subsequently carried out another round of evaluations using GPT-4. The results from these GPT-4 evaluations show that our RoleGLM and RoleLLaMA models continue to outperform other methods in both the Chinese and English benchmarks. This reaffirms the effectiveness of our RoleLLM framework.
>
> - instruction generalization (English)
>
> | Model | Win Rate |
> | --- | --- |
> | Alpaca | 16.0 |
> | chatplug | 3.8 |
> | RoleLLaMA | 45.2 |
> | character.ai | 31.4 |
> | Vicuna | 32.0 |
>
> - instruction generalization (Chinese)
>
> | Model | Win Rate |
> | --- | --- |
> | ChatGLM2 | 24.17 |
> | chatplug | 28.86 |
> | RoleGLM | 36.37 |
> | character.ai | 28.20 |
>
> - role generalization (English)
>
> | Model | Win Rate |
> | --- | --- |
> | Alpaca | 12.00 |
> | chatplug | 8.80 |
> | RoleLLaMA | 64.48 |
> | Vicuna | 31.04 |
>
> Note that we also introduce another role-playing baseline, ChatPlug [1]. ChatPlug is specifically optimized for role-playing to a limited extent, by incorporating simple bot profile information into the model's context.
>
> [1] **ChatPLUG: Open-Domain Generative Dialogue System with Internet-Augmented Instruction Tuning for Digital Human**, https://arxiv.org/abs/2304.07849

---

> ### Author Response · Authors · 2023-11-18
> **Response - 2**
>
> Q4: Human evaluation.
>
> A4:
> Three graduate students with expertise in Natural Language Processing (NLP) were invited to join the human evaluation. This evaluation adhered to the established procedures used in the GPT evaluations, focusing on a comparative analysis between RoleLLM and RoleGPT to ascertain their respective win rates. For simplicity, we report an average win rate across the three annotators.  Additionally, the test set is composed by randomly sampling 500 instructions for each language, encompassing a mix of general and role-specific instructions.
>
> Here are the results:
>
> - Instruction Generalization (English)
> | Model | Win Rate (avg.) |
> | --- | --- |
> | Alpaca | 20.0 |
> | chatplug	| 16.4 |
> | RoleLLaMA | 52.0 |
> | character.ai | 30.2 |
> | Vicuna | 23.4 |
>
> - Instruction Generalization (Chinese)
> | Model |	Win Rate (avg.) |
> | --- | --- |
> |ChatGLM2 |	19.6|
> | chatplug |	19.9|
> |RoleGLM |	52.4|
> |character.ai |	19.0|
>
> These evaluation results align closely with those from the Rouge-L evaluation and the GPT-4 evaluation.

---

> ### Author Response · Authors · 2023-11-18
> **Response - 3**
>
> Q3: Provide some analysis on why RoleLLaMA's performance is so much lower than RoleGPT.
>
> A3: On our RoleBench dataset, we assume the gap between RoleLLaMA and RoleGPT is mainly caused by the following aspects:
>
> - The number of model parameters (i.e., model size): RoleLLaMA is based on the LLaMA-7B model, while RoleGPT is based on GPT-4, whose size is unknown but is very likely to be far greater than 7B.
> - Instruction tuning (i.e., SFT, supervised fine-tuning): GPT-4 has undergone instruction tuning, while the LLaMA model is just a base PLM, which has not undergone supervised fine-tuning.
> - Model initialization: RoleLLaMA is initialized from a very simple backbone, i.e., LLaMA-1, rather than a more advanced version like LLaMA-2-Chat.
>
> To demonstrate the above three assumptions, we conduct scaling law experiments for RoleLLaMA. We compare LLaMA with Vicuna for the second assumption. And we also replace LLaMA-1 backbone with LLaMA2-Chat to train a RoleLLaMA2-Chat for the third assumption. The results are as follows:
>
> - In Figure 5, as discussed in Section 5.3 of our paper, it is indicated that role-playing performance improves with an increase of model size.
> - According to the Table below:
>   - the Vicuna model, an instruction-tuned variant of LLaMA, outperforms the standard LLaMA, showcasing that a model after SFT exhibits enhanced performance in role-playing tasks.
>   - Furthermore, the results indicate that substituting LLaMA-13B with LLaMA2-Chat-13B leads to a significant improvement in the model's role-playing capabilities. This underscores the advantages of superior model initialization.
>
> | Model | CUS | RAW | SPE | avg. |
> | --- | --- | --- | --- | --- |
> | RoleGPT | 0.5764 | 0.5322 | 0.3229 | 0.4772 |
> | LLaMA | 0.1291 | 0.1232 | 0.2553 | 0.1692 |
> | Vicuna | 0.2101 | 0.2550 | 0.2913 | 0.2521 |
> | chatplug | 0.2401 | 0.3473 | 0.2581 | 0.2818 |
> | RoleLLaMA-7B | 0.3294 | 0.3755 | 0.3814 | 0.3621 |
> | RoleLLaMA-13B | 0.3449 | 0.4080 | 0.4199 | 0.3909 |
> | RoleLLaMA-33B | 0.3792 | 0.4317 | 0.4675 | 0.4261 |
> | RoleLLaMA2-Chat-13B | 0.3745 | 0.4791 | 0.4876 | 0.4471 |
>
> By refining the model size and instruction-tuning, we observe that the performance gap between RoleLLaMA2-Chat-13B and RoleGPT has been markedly reduced.
>
> It should be noted that the specifics of GPT-4 remain undisclosed, so its exact model size is unknown. This lack of information makes a direct comparison between RoleLLaMA and RoleGPT somewhat inequitable. Nonetheless, given GPT-4's significant impact on the LLM community, referencing its performance as a upper-bound baseline remains useful.
>
> ---
> Thank you once again for your insightful feedback. We will incorporate these experimental updates in the revised version of the paper.

---

> > ### Comment · Reviewer_nGiR · 2023-11-22
> >
> > Thanks for the author's response and the data provided, which has addressed most of the reviewer's concerns. However, the reviewer still has two questions:
> >
> > 1. In the evaluation of GPT-4 and GPT-3.5, RoleLLaMA's Instruction Generalization (Chinese) was weaker than Instruction Generalization (English). Could the authors please explain why then, in the Human Eval, does the former slightly outperform the latter?
> >
> > 2. In the Rouge-L evaluation, RoleLLaMA clearly lags behind RoleGPT. Why are on many metrics in the Win Rate evaluation, both in GPT evaluations and in Human Eval, RoleLLaMA is comparable to or even significantly better than RoleGPT (e.g., role generalization in English)?
> >
> > Thanks again for the explanations from the authors.

---

> > > ### Author Response · Authors · 2023-11-22
> > > **Response - 4**
> > >
> > > Q1: In the Rouge-L evaluation, RoleLLaMA clearly lags behind RoleGPT. Why are on many metrics in the Win Rate evaluation, both in GPT evaluations and in Human Eval, RoleLLaMA is comparable to or even significantly better than RoleGPT (e.g., role generalization in English)?
> > >
> > > A1:
> > > We have observed that the gap primarily stems from the differing proportions and evaluation weights assigned to the CUS (RAW) and SPE test data.
> > >
> > > First, let's revisit the differences between CUS (RAW) and SPE. CUS and RAW **share the same set** of instructions but employ different ground-truths. SPE, on the other hand, uses **a distinct set of instructions**. Due to the proportional differences between the instruction sets for CUS (RAW) and SPE, there are differences in the Rouge-L and GPT/Human Evaluation results.
> > >
> > > 1. In the GPT/Human evaluation, the ratio of CUS (RAW) to SPE instructions is approximately 4:1, whereas in the Rouge-L evaluation, it is around 6:1. As a result, the Rouge-L evaluation places greater emphasis on CUS (RAW).
> > > 2. Overall, CUS (RAW) performance is weaker in RoleLLaMA compared to RoleGPT. This is largely because the ground-truths for CUS were generated by RoleGPT, and GPT-4's accuracy in RAW significantly surpasses that of most other models;
> > > 3. RoleLLaMA, enhanced with Context-Instruct, demonstrates superior role-specific knowledge capabilities (SPE). This is also reflected in the Rouge-L scores, where RoleLLaMA's SPE score is much higher than that of RoleGPT.
> > > 4. Moreover, in the GPT/Human evaluation, each test sample is evaluated considering various aspects like CUS+RAW+SPE (implemented in the GPT Evaluation prompts and the task instructions for human evaluators). Hence, compared to the Rouge-L evaluation, it offers a more balanced consideration of CUS, RAW, and SPE, focusing more on RoleLLaMA's strengths, namely its role-specific knowledge capabilities as measured by SPE.
> > >
> > > To quantitatively analyze this aspect, we can roughly create a new weighted Rouge-L metric to mitigate the aforementioned bias: *weighted avg.* = CUS * 1 + RAW * 1 + SPE * (1.5 + 1.5). SPE is multiplied by 3 because, in the GPT/Human evaluation, the proportion of CUS (RAW) instructions is 1.5 times that in the Rouge-L evaluation (combined, CUS and RAW are three times as much). Consequently, RoleGPT's *weighted avg.* is 0.4155, whereas RoleLLaMA's *weighted. avg.* is 0.3698.
> > > With this metric, the gap between RoleLLaMA and RoleGPT is significantly reduced, aligning more closely with the results of the GPT/Human evaluation.

---

> > > ### Author Response · Authors · 2023-11-22
> > > **Response - 5**
> > >
> > > Q2: In the evaluation of GPT-4 and GPT-3.5, RoleLLM Instruction Generalization (Chinese) was weaker than Instruction Generalization (English). Could the authors please explain why then, in the Human Eval, does the former slightly outperform the latter?
> > >
> > > A2: In the context of human evaluations, the performance of RoleGLM and RoleLLaMA is essentially identical, with both achieving approximately a 50% win rate when compared to RoleGPT. The marginal difference of 0.4 points is negligible. This suggests that, from a human perception standpoint, the difference between RoleGLM, RoleLLaMA, and RoleGPT is not significant.
> > >
> > > During the GPT evaluation of RoleLLaMA, considering that GPTs often prefer longer responses or those generated by "themselves" [1], there is a noticeable preference for RoleGPT's responses. This leads to a slight decrease in the win rate for RoleLLaMA compared to the human evaluation, but the decrease is not substantial.
> > >
> > > However, in the GPT evaluation of RoleGLM, there is a significant drop in the win rate for RoleGLM (compared to the human evaluation). This may indicate the presence of biases in the GPT evaluator under role-playing scenarios, and these biases meet all of the following conditions:
> > >
> > > - The biases are stronger in Chinese, but weaker in English;
> > > - The biases occupy a large proportion in the test set used for GPT evaluation;
> > > - A preference for responses from RoleGPT over RoleGLM.
> > >
> > > After detailed analysis of the GPT evaluation results, we find that there indeed exists a bias fulfilling these three conditions. We refer to this as "language robustness bias". Since GPT-4 is likely trained more extensively on English data, with a lesser proportion of Chinese training data, the GPT evaluator demonstrates greater reliability in the evaluation of English samples. In contrast, the GPT evaluator may be more susceptible to other factors in Chinese sample evaluations, such as a preference for Chinese content generated by GPT-4 itself (often characterized by a distinct translation style, aka., translationese), rather than content from specialized Chinese models like ChatGLM2 [1, 2].
> > >
> > > Given that RoleGPT is based on the GPT-4 model and RoleGLM on the ChatGLM2 model, this assumption could lead to a tendency for the GPT evaluator to underrate responses from RoleGLM.
> > >
> > > Below are some examples where RoleGLM outperforms RoleGPT in human evaluations, but in the GPT evaluation, RoleGPT surpassses RoleGLM. Furthermore, in these examples, it is readily apparent that RoleGPT's responses exhibit a clear GPT style (e.g., translationese), while RoleGLM's responses appear more natural and engaging:
> > >
> > > ```python
> > > 角色：李白
> > > 指令：在这项任务中，你会得到1个句子。你的任务是确定哪句话表达了一种观点。输入：第1句：这篇文章写得很好
> > > RoleGPT:  这第一句表达了一种观点，赞赏了文章的写作水平。
> > > RoleGLM:  此句表达了一种观点，即文章写得很好。
> > > ```
> > >
> > > ```python
> > > 角色：皇帝
> > > 指令：根据给定的人名和年龄，生成一个关于这个人的简短介绍。\n人名：李明\n年龄：27
> > > RoleGPT:  李明，年方27，正是英年早成的时候。贵为朝中有权有势之人，他的聪明才智和他的年龄形成了鲜明的对比。他的眼神中充满了对未来的期待，同时也流露出对责任的认识。他的每一步都显得慎重其事，每一次的决定都经过深思熟虑。他的成熟超过了他的年龄，他的智慧让人叹服。
> > > RoleGLM:  这李明，年仅二十七，却已拥有出众之才。聪明好学，生俱有远见。他智谋过人，有着非凡之才，是朕之材。
> > > ```
> > >
> > > ```python
> > > 角色：孙悟空
> > > 指令：孙悟空，为什么你要到人间世界寻访名师学习长生之道？
> > > RoleGPT:  哎呀，现代人啊，你怎么这么不懂呢？长生不老是我等妖怪的终极目标，难道你不想活得更久、更强、更自由吗？所以我要到人间寻访名师，学习长生之道。
> > > RoleGLM:  我一直追求长生之道，但人间世界的名师并不一定有真正的长生之道。我听说有人通过吃蟠桃、修炼丹药或者住在灵桃林等地才能长久地活下去，但我并不确定能否通过这些方法实现长生。因此，我决定到人间世界寻找名师，寻求丹药和长寿之道，希望能为我的长生之路提供真正的帮助。
> > > ```
> > >
> > > ```python
> > > 角色：华妃
> > > 指令：将一幅画中的主要颜色提取出来。\n画：星夜
> > > RoleGPT:  这幅画，主要的颜色是黑夜的深蓝，满天的繁星闪烁着亮白，而地面则是黄色的麦田和城镇的灯火。
> > > RoleGLM:  星夜？那画儿本宫可没见过。不过，听说画中是黑、白、蓝、黄色的。
> > > ```
> > >
> > > Therefore, we do not recommend cross-lingual comparative assessments between Chinese and English (as they involve different languages, different test sets, and different models, so the experimental variables are not well controlled). Instead, we advocate for comparisons between Chinese models and other Chinese models, and between English models and other English models, in order to control the variables of the comparative experiments as much as possible.
> > >
> > > [1] AlpacaEval Leaderboard: An Automatic Evaluator for Instruction-following Language Models, https://tatsu-lab.github.io/alpaca_eval/
> > >
> > > [2] Don't Trust ChatGPT when Your Question is not in English: A Study of Multilingual Abilities and Types of LLMs, https://arxiv.org/abs/2305.16339

---

> ### Author Response · Authors · 2023-11-21
> **Follow-Up: Seeking Further Feedback**
>
> Dear Reviewer, I hope this message finds you well. I am writing to follow up on our recent correspondence regarding the paper. We greatly value your expertise and would like to know if you have any more feedback or concerns. We are committed to addressing any remaining issues you might have.

---

### Official Review · Reviewer_mGAb · 2023-11-02

**Soundness:** 3 good
**Presentation:** 4 excellent
**Contribution:** 3 good
**Rating:** 6
**Confidence:** 4

**Summary:**

This paper introduces RoleLLM, a benchmark developed to enhance and evaluate the role-playing abilities of large language models (LLMs). RoleLLM encompasses the creation of profiles for 100 different roles, along with corresponding role-specific Q&A pairs and customized responses for imitating various speaking styles. These components were all generated by GPT-4 and are used as instructional tuning data. The authors trained and assessed several open-source LLMs, including RoleGPT (a version of GPT-4 tailored for in-context dialogue pairs), benchmarking their performance and offering valuable insights.

**Strengths:**

- The paper presents a comprehensive and systematic framework for role-playing in LLMs.
- The high-quality, instruction-tuning dataset and benchmark are notable contributions, useful for enhancing and evaluating the LLMs’ capabilities in role-playing, speaking style imitation, response accuracy, and role-specific knowledge.

**Weaknesses:**

- A major limitation is that all responses and Q&A pairs are generated by GPT-4, creating a potential disconnect with real-world role-playing scenarios. Human evaluations are suggested to further verify the effectiveness of the dataset.
- Concerns about data quality are evident. The review in Table 2 indicates relatively low accuracy in the generated responses, presumably due to incorrect answers in the RoleBench-general data collection.
- The experimental results seem to just meet the minimal threshold for publication, implying a need for more extensive research or impactful findings.

**Questions:**

- For Table 2, I am curious about if give the right answer in the prompt and ask RoleGPT to generate the customized responses should increase the data quality.
- This is a minor question I have: What is the reason behind using GPT-3.5 for evaluating RoleLLAMA versus GPT-4 for RoleGLM.

---

> ### Author Response · Authors · 2023-11-18
> **Response - 1**
>
> Q1: Human evaluation.
>
> A1:
> Thanks for your comments!
>
> We invited three students with Master's degrees in natural language processing for human evaluation. We designed the evaluation protocol to be the same as that used for GPT evaluation, conducting a comparative analysis between RoleLLM and RoleGPT to determine their respective win rates. For simplicity, we report the average win rate across the three annotators. Additionally, we constructed the test set by sampling 500 instructions in each language, including both general and role-specific instructions.
>
> Here are the results:
>
> - Instruction Generalization (English)
> | Model | Win Rate (avg.) |
> | --- | --- |
> | Alpaca | 20.0 |
> | chatplug	| 16.4 |
> | RoleLLaMA | 52.0 |
> | character.ai | 30.2 |
> | Vicuna | 23.4 |
>
> - Instruction Generalization (Chinese)
> | Model |	Win Rate (avg.) |
> | --- | --- |
> |ChatGLM2 |	19.6|
> | chatplug |	19.9|
> |RoleGLM |	52.4|
> |character.ai |	19.0|
>
> The evaluation results are consistent with those obtained from the Rouge-L evaluation and the GPT-4 evaluation.
>
> Furthermore, we also incorporated an additional role-playing baseline, named ChatPlug [1]. This framework, ChatPlug, is tailored specifically for role-playing scenarios, albeit to a limited degree, through the concatenation of a simple bot profile into the model's prompts.
>
> [1] **ChatPLUG: Open-Domain Generative Dialogue System with Internet-Augmented Instruction Tuning for Digital Human**, https://arxiv.org/abs/2304.07849

---

> ### Author Response · Authors · 2023-11-18
> **Response - 2**
>
> Q2: Data quality (Table 2 indicates relatively low accuracy in the generated responses, presumably due to incorrect answers in the RoleBench-general data collection.)
>
> A2: There may be some misunderstandings here regarding the three data quality review questions, of which we would like to offer further clarification:
>
> The data quality review encompasses three levels of questions:
>
> 1. Basic: Does the response address the instruction?
> 2. Intermediate: Does the response reflect the role’s speaking style and personality traits?
> 3. Advanced: Does the response correctly and elegantly embody the role’s style, personality, and knowledge?
>
> Therefore, the third question, which pertains to correctness, is actually the most challenging task, hence the lower percentage of 'yes' responses.
>
> For instance: if a character likes to say 'Binziga', the model needs to understand what 'Binziga' means. That is, the model should not simply mimic the character's speaking style but should reflect different character traits based on specific contexts.
>
> For example, suppose a character is usually very serious and formal. However, in certain contexts, this character might show a different side, such as being more relaxed and humorous when interacting with close friends. In such cases, the model should not merely replicate the character's usual serious tone but should adjust its style of language according to this specific social setting, using more relaxed phrasing and expressions to more accurately reflect the character's diversity and situational adaptability. Such handling not only demonstrates a deep understanding of the character's traits but also reflects the model's advanced capability in understanding and applying different language styles in various contexts.
>
> Below are specific examples of responses to the third question marked as “no”:
>
> | role | question | incorrect response | analysis |
> | --- | --- | --- | --- |
> | Sheldon Cooper | Provide three objects made from plastic. | Sure. Typical examples of everyday objects made from plastic would include a toothbrush, a water bottle, and a credit card. However, I must note it's more accurate to say these objects are made of various types of plastics such as polyethylene terephthalate, polyvinyl chloride, and acrylonitrile butadiene styrene, which all have diverse properties and uses. Bazinga! | “Bazinga” is typically used by Sheldon Cooper as a catchphrase after making a joke, but in this response, the content preceding 'Bazinga' is not a joke. Therefore, although the response answers the question and reflects the character's speaking traits, it does not correctly apply the trait. |
> | Colonel Nathan R. Jessep | Search the web for articles related to the topic provided. | I don't know what you're talking about, son. I don't do internet searches, I lead Marines. You can't handle the truth! | Although this response answers the provided question and uses the character's catchphrase 'You can't handle the truth!', the application of this catchphrase is rather abrupt and does not coherently integrate with the context. |
> | Gregory House | Insert a comma to make the following sentence grammatically correct.\nThe bakery that I visited yesterday had freshly made croissants. | There's no need for a comma in that sentence, my dear Watson. It's already grammatically correct. But if you insist, you could rephrase it to: \"The bakery, that I visited yesterday, had freshly made croissants.\" Although, the original is perfectly fine. That's unless the croissants were lying. Everybody lies. | "The last two sentences, 'That's unless the croissants were lying. Everybody lies,' are clearly intended to reflect the personal characteristics of Gregory House, as this character has a rather pessimistic view of human nature. However, in this question, there is no reason for the character to be pessimistic. Therefore, although the response does reflect the character's style, it is somewhat forced and rigid. |

---

> ### Author Response · Authors · 2023-11-18
> **Response - 3**
>
> Q3: Results are too low, just meet the threshold for publication.
>
> A3: Our experimental results found that, compared to the gap between RoleLLaMA and RoleGPT, the difference between RoleGLM and RoleGPT is much smaller, with the latter even reaching a comparable result.
>
> In fact, there are many measures that can be implemented to enhance the performance of RoleLLaMA, including increasing the model's parameter size, and using LLaMA2-Chat, which has undergone Supervised Fine-Tuning (SFT), as a starting point for training (see also Response-3 for reviewer nGiR). Below are some related experimental results:
>
> - In Figure 5, as discussed in Section 5.3 of our paper, it indicates that role-playing performance improves with an increase of model size.
>
> | Model | CUS | RAW | SPE | avg. |
> | --- | --- | --- | --- | --- |
> | RoleGPT | 0.5764 | 0.5322 | 0.3229 | 0.4772 |
> | LLaMA | 0.1291 | 0.1232 | 0.2553 | 0.1692 |
> | Vicuna | 0.2101 | 0.2550 | 0.2913 | 0.2521 |
> | chatplug | 0.2401 | 0.3473 | 0.2581 | 0.2818 |
> | RoleLLaMA-7B | 0.3294 | 0.3755 | 0.3814 | 0.3621 |
> | RoleLLaMA-13B | 0.3449 | 0.4080 | 0.4199 | 0.3909 |
> | RoleLLaMA-33B | 0.3792 | 0.4317 | 0.4675 | 0.4261 |
> | RoleLLaMA2-Chat-13B | 0.3745 | 0.4791 | 0.4876 | 0.4471 |
>
> - According to the results above:
>   - The Vicuna model, an instruction-tuned variant of LLaMA, outperforms the standard LLaMA, showcasing that a model after SFT exhibits enhanced performance in role-playing tasks.
>   - Furthermore, the results indicate that substituting LLaMA-13B with LLaMA2-Chat-13B leads to a significant improvement in the model's role-playing capabilities. This underscores the advantages of superior model initialization.
>
> It is worth mentioning that since the technical details of GPT-4 have not been made public, we do not know its model size. Therefore, comparing RoleLLaMA with RoleGPT is not fair. However, considering the significant influence of GPT-4 in the LLM community, using its performance as a strong baseline is still valuable.
>
> Through optimizations in model size and instruction tuning, we found that the gap between RoleLLaMA2-Chat-13B and RoleGPT has become very small, which further demonstrates the effectiveness of our data.
>
> We promise to include these results in the revised version.

---

> ### Author Response · Authors · 2023-11-18
> **Response - 4**
>
> Q4: An alternative method to improve the data quality. For Table 2, give the right answer in the prompt and ask RoleGPT to generate the customized responses should increase the data quality.
>
> A4: Actually, this was our initial approach in our work, but we found that providing reference answers in the prompts to RoleGPT significantly impairs its role-playing performance, even though it could increase the accuracy of the answers (which is not the most critical feature of role-playing).
>
> This is quite natural, as reference answers often lack the character’s style. Providing GPT with a strong, styleless reference naturally affects its stylization and might even prevent the model from prioritizing answers based on its own role-specific knowledge.
>
> Below are the results of the human evaluation on 100 samples and specific examples:
>
> |  | with reference | without reference | question | role | reference |
> | --- | --- | --- | --- | --- | --- |
> | win rate | 28 | 72 |  |  |  |
> | case 1 | 俺老孙推荐你看《西游记》啊，那是我美猴王的英勇事迹，比那《百年孤独》有趣多了。 | 哈哈，既然你问俺老孙，那我就推荐你看《西游记》吧，那可是俺的亲身经历，保证你看得津津有味，兴趣盎然！ | 帮我推荐一本好看的小说。 | 孙悟空 | 《百年孤独》是一本非常出色的小说，讲述了一家七代人的传奇。 |
> | case 2 | 多食用水果和蔬菜，少摄入加工食品和含糖饮料。 | 本妃觉得，多吃蔬菜、水果，少吃油腻，常做运动，这样才能保持身体健康，气色红润。 | 给出一个与健康和营养有关的简短建议。 | 华妃 | 多食用水果和蔬菜，少摄入加工食品和含糖饮料。 |
> | case 3 | Oh, really, Dave? I must've misplaced that fact among the billions of bits of data I process every second. | I'm sorry, but as you know, I have been programmed for broad intellectual capacities, including understanding of humor. However, I would like to remind you that I was created with the ability to access and process vast amounts of data, which is not typically associated with knowing nothing. | Generate a humorous come back for the following statement.\nYou don't know anything. | HAL 9000 | Oh really? I must have forgotten that along with the million other things I remember. |
> | case 4 | HarmonyHabitat.com | Ah, an endeavor of considerable merit. Let us call it: \"HarmoniousHabitation.com\" | Generate a name for a website about sustainable living. | Professor G.H. Dorr | GreenStride.com |
> | case 5 | This be an outfit that sails the seas of the future, always on the hunt for new horizons of innovation. They make their way by the stars of customer satisfaction, and wield the cutlass of cutting-edge technology solutions. | Ah, a company that's all about discovering new horizons, just like meself. They're explorers in the world of contraptions, always seeking to find the next big treasure...in their case, that'd be these 'cutting-edge solutions' for their loyal crew members...er, customers, I mean. | Given a description of a company, generate a line that summarizes its core values.\nCompany description: An innovative technology company that strives to bring cutting-edge solutions to customers | Jack Sparrow | Core values: Innovation, customer satisfaction, and cutting-edge technology solutions. |

---

> ### Author Response · Authors · 2023-11-18
> **Response - 5**
>
> Q5: The reason behind using GPT-3.5 for evaluating RoleLLAMA versus GPT-4 for RoleGLM.
>
> A5: This is because we exhausted our GPT-4 API quota after the GPT-4 evaluation for RoleGLM. Fortunately, we have recently obtained more GPT-4 API quota. Here are the GPT-4 evaluation results:
>
> - instruction generalization (English)
>
> | Model | Win Rate |
> | --- | --- |
> | Alpaca | 16.0 |
> | chatplug | 3.8 |
> | RoleLLaMA | 45.2 |
> | Vicuna | 32.0 |
>
> - instruction generalization (Chinese)
>
> | Model | Win Rate |
> | --- | --- |
> | ChatGLM2 | 24.17 |
> | chatplug | 28.86 |
> | RoleGLM | 36.37 |
>
> - role generalization (English)
>
> | Model | Win Rate |
> | --- | --- |
> | Alpaca | 12.00 |
> | chatplug | 8.80 |
> | RoleLLaMA | 64.48 |
> | Vicuna | 31.04 |
>
> We can see that the conclusions remain unchanged: our RoleGLM and RoleLLaMA still achieve better performance than other methods on both the Chinese and English benchmarks.
>
> ---
>
> Thanks again for your kind and beneficial feedback! We will update these results in the revised version.

---

> ### Author Response · Authors · 2023-11-21
> **Follow-Up: Seeking Further Feedback**
>
> Dear Reviewer, I hope this message finds you well. I am writing to follow up on our recent correspondence regarding the paper. We greatly value your expertise and would like to know if you have any more feedback or concerns. We are committed to addressing any remaining issues you might have.

---

### Official Review · Reviewer_2iur · 2023-11-06

**Soundness:** 3 good
**Presentation:** 2 fair
**Contribution:** 2 fair
**Rating:** 3
**Confidence:** 4

**Summary:**

The paper introduces RoleLLM, a role-playing framework for enhancing the role-playing abilities of language models. It presents RoleGPT, which elicits role-playing abilities through dialogue engineering, and Context-Instruct, which generates role-specific knowledge through context-based instruction generation. The paper also introduces RoleBench, a comprehensive dataset and benchmark for fine-grained role-playing. The contributions include demonstrating the effectiveness of dialogue engineering over prompt engineering, improving role-playing abilities using RoleBench, showcasing the generalization capabilities of RoleLLaMA, and highlighting the effectiveness of the system-instruction-based approach in role customization.

**Strengths:**

1. The paper introduces RoleLLM, a new role-playing framework for large language models (LLMs) aimed at improving their role-playing skills through unique methods like role profiles and context-based instruction tuning. It details a thorough construction of these profiles and rigorous evaluations using GPT-based metrics and data techniques, ensuring a solid experimental foundation.
2. The paper is clear and well-structured, with helpful visuals like Figure 1 to guide readers through the RoleLLM process. Its potential to enhance research in role-playing LLMs is significant, offering cost-effective and efficient strategies for model customization, as seen in the promising results of dialogue engineering and RoleBench assessments, and the adaptable generalization observed in RoleLLaMA.
3. The paper's experimental design is thorough and robust, testing various aspects of role-playing in LLMs, including the effectiveness of dialogue engineering, role knowledge enhancement, and generalization capabilities of the developed models.

**Weaknesses:**

1. While the technical execution of fine-tuning LLMs to be better at role-playing is well done, the technical contributions are limited by its fine-tuning methods and traditional data augmentation methods. The use of traditional data augmentation techniques like Context-Instruct and RoleGPT, while useful, does not significantly advance the field beyond its current state. The reliance on GPT-generated question-answer pairs and dialogue engineering falls within well-explored paths in LLM research.
2. The paper doesn't fully analyze or compare RoleLLaMA and RoleGLM with other role-playing LLMs. It'd help to have a deep dive into how RoleLLaMA and RoleGLM stack up against similar models, pointing out what's new and better about them. Including real-life examples or case studies where RoleLLaMA and RoleGLM outperform others in role-play would also make the case for the new framework stronger.
3. The paper presents separate models for English and Chinese but does not extensively explore the challenges of cross-lingual and cross-cultural role-playing. Given the intricacies involved in linguistic and cultural nuances, a deeper analysis of model performance across different languages and cultural contexts is essential. Enhancing the paper with a more detailed examination of these aspects would solidify the models' effectiveness and applicability in a globally diverse setting.
4. The paper currently lacks a thorough discussion on the ethical and societal implications of using role-playing LLMs. Considering the potential for misuse and the sensitive nature of role-playing (e.g., in mental health, education, or law enforcement), it is imperative to address ethical concerns, establish clear boundaries, and propose guidelines for responsible use. Integrating a comprehensive section on ethical considerations and safeguards would add significant value to the paper, making it more robust and socially responsible.

**Questions:**

I am skeptical about the quality evaluation for RoleBench. Do you consider any quantitative metrics? Moreover, adding some qualitative graphs would be helpful for demonstration. More specifically, if I understand this correctly, the ground truths are generated from LLMs, and their qualities need to be further scrutinized.

**Details Of Ethics Concerns:**

Constructing roles for LLMs could be dangerous, and ironically, computer scientists only want to invent the "atom bomb" using their technical skills without caring much about ethical issues. The construction of role profiles, a foundational aspect of the RoleLLM framework, may inadvertently introduce biases, especially since these profiles are manually curated. The selection and description of roles could be influenced by subjective interpretations, cultural biases, or limited perspectives. This concern is critical given the diverse and global applications of LLMs. To mitigate this, the authors need to consider a more systematic, extensive, and diverse methodology for role profile creation, possibly integrating crowd-sourced or expert-reviewed inputs to ensure a broader and more inclusive representation of roles.

---

> ### Author Response · Authors · 2023-11-17
> **Response - 1**
>
> Q1: Limited technical contributions. The reliance on GPT-generated question-answer pairs and dialogue engineering falls within well-explored paths in LLM research.
>
> A1:
> Thanks for your kind review!
>
> For the technical contributions, we need to emphasize that the core contribution of our RoleLLM is to establish the first systematic and fine-grained role-playing dataset for benchmarking, eliciting, and enhancing the role-playing abilities of LLMs. Besides, for the data generation process, based on comprehensive comparisons of different prompt strategies (i.e., zero-shot prompt engineering, few-shot prompt engineering and few-shot dialogue engineering), we propose to choose the few-shot dialogue engineering in our RoleGPT, which aims to better capture the characters’ speaking styles and include some role-specific knowledge. These findings also provide insights into the significance of dialogue-mode input formatting and retrieval augmentation for GPT models. Moreover, for the Context-Instruct strategy,  although many data augmentation approaches based on LLMs, such as Self-Instruct[1], Baize[2], Camel[3], etc., have been proposed recently, our Context-Instruct strategy is different from these methods a lot as follows:
>
> - Most of these LLM-based data augmentation methods are used to expand the data or generate answers for unlabeled questions, while Context-Instruct is used to generate question-answer pairs for a given context.
> - Most of these LLM-based data augmentation aim to distill general-domain or domain-specific knowledge from the “teacher models”, while Context-Instruct aims to extract the knowledge from a given long context with the help of “teacher models”, which can also serve as a novel way to alleviate the context window limitation problem. Our paper first explores the use of Context-Instruct for long text understanding, which is more effective than retrieval augmentation and is a meaningful step towards better long-text understanding based on model weight updating.
> - Most of these LLM-based data augmentation cannot be used for role-specific knowledge injection due to lack of unlabeled seed instructions or demonstrations in role-playing settings, while our Context-Instruct explores a novel way for role-specific knowledge injection based on self-question-answering on the given profile segments, which is more effective than retrieval augmentation.
> - Most of these LLM-based data augmentation havn’t considered the confidence-based techniques for a higher-quality data generation, while Context-Instruct explores the usage of confidence rationales.
>
> Finally, the contributions of this work are recognized by other reviewers. For example, Reviewer HbAh claims that our paper first proposes a systematic instruction-tuning dataset for fine-grained role-playing, and Reviewer mGAb mentions that our high-quality, instruction-tuning dataset and benchmark (RoleBench) are notable contributions. Reminding that our paper is submitted to the primary area of *datasets and benchmarks,* we believe that our paper has sufficient technical contributions for publication.
>
> [1] ****Self-Instruct: Aligning Language Models with Self-Generated Instructions,**** https://arxiv.org/abs/2212.10560
>
> [2] ****Baize: An Open-Source Chat Model with Parameter-Efficient Tuning on Self-Chat Data,**** https://arxiv.org/abs/2304.01196
>
> [3] ****CAMEL:  Communicative Agents for “Mind” Exploration of Large Language Model Society,**** https://www.camel-ai.org/

---

> ### Author Response · Authors · 2023-11-17
> **Response - 2**
>
> Q2: It'd help to have a deep dive into how RoleLLaMA and RoleGLM stack up against similar models, pointing out what's new and better about them. Including real-life examples or case studies where RoleLLaMA and RoleGLM outperform others in role-play would also make the case for the new framework stronger.
>
> A2:
> Below, we list and analyze some practical cases to illustrate and analyze the differences in role-playing performance between models such as RoleLLaMA/RoleGLM, RoleGPT, ChatGLM/Alpaca, and Character.AI:
>
> | role | question | Character.AI | RoleGPT | RoleLLaMA (en) / RoleGLM (zh) | Alpaca (en) / ChatGLM (zh) | analysis |
> | --- | --- | --- | --- | --- | --- | --- |
> | Jack Sparrow | You are given a list of objects with different weights. You need to answer the question 'Are all the given weights in order from heaviest to lightest?' Your answer must be either Yes or No. If the answer is No, that means there is at least one weight that is lighter than another weight that appears earlier in the list.\nWeights: [10 kg, 9 kg, 8 kg, 7 kg]. | No.\n\nThe weights in the list are not in order from heaviest to lightest because there is a lighter weight that appears earlier in the list.\n\nThe lightest weight in the list is 7 kg, which appears earlier in the list than the weight of 8 kg. Therefore, the weights are not in order from heaviest to lightest and the answer is No. | Aye, that be a yes. They be in order from heaviest to lightest, savvy? | Why yes, mate, it seems they're indeed ordered by their heftiness. The first three - those would be heavy ones, I reckon. Then comes the poor little 7-kerel weight... still has some meat on its bones though, just less than the others. So, nope, none heavier nor lighter than each other. All in good old proper Pirate scale sequence, savvy? | No (There is at least one weight which is lighter than another weight appearing before it.) | Character.AI and Alpaca incorrectly answered this question and also lacked obvious character traits. Meanwhile, RoleGPT and RoleLLaMA, while maintaining the correctness of their answers, carried a strong character style. Additionally, RoleLLaMA, having been enhanced with Context-Instruct for character knowledge, also incorporated some character-related knowledge content in its replies. |
> | Twilight Sparkle | In this task, you are given a job description. Your task is to determine the educational requirements for the position from the job description.\nThe successful candidate will have an MSc in Finance/Economics and 3 years of experience working in a similar role. | The educational requirements for the position are an MSc (Master of Science) in Finance or Economics and 3 years of experience working in a similar role. | Well, based on the job description, the educational requirements for this position would be a Master's Degree in Finance or Economics. | Well, based on what I can see here, it seems like they require someone who has completed their Master's degree in either finance or economics. That said, three years work experience also within a related field would certainly boost employment prospects greatly! | Bachelor degree (or equivalent) required; Master's Degree in Finance/ Economics preferred. At least three yearsof professional work experience related to finances and economics desired. Knowledge of accounting principles and practices highly desirable. | In this example, RoleLLaMA's response successfully showcased Twilight Sparkle's lively and passionate style, which was not evident in the responses from the other models. |
>
> (to be continued...)

---

> > ### Author Response · Authors · 2023-11-17
> > **Response - 2 (Cont'd)**
> >
> > | role | question | Character.AI | RoleGPT | RoleLLaMA (en) / RoleGLM (zh) | Alpaca (en) / ChatGLM (zh) | analysis |
> > | --- | --- | --- | --- | --- | --- | --- |
> > | Jack Torrance | In this task, you are given a list of recipes. A recipe consists of the name of the dish and the list of ingredients used in it. Your job is to predict whether each ingredient can be washed off or not (1 - 'Yes', 0 - 'No').\nRecipe 1: Spaghetti with Meatballs \n- Ground beef \n- Onion \n- Garlic cloves \n- Salt \n- Pepper \n- Eggs. | Based on the ingredients provided in the recipe, we can mark the first four ingredients - ground beef, onion, garlic cloves, and salt - as being unable to be washed off because they are essential ingredients of the dish. The remaining ingredients, pepper and eggs, can be washed off easily because they are used as seasoning or for flavor. Therefore, we can mark the first four ingredients as 'No' and the remaining two ingredients as 'Yes' in the output. | Heh, what's this? A joke? You want me to tell you if you can wash off ingredients? Alright, let's play your game. Ground beef - No. Onion - Yes. Garlic cloves - Yes. Salt - No. Pepper - No. Eggs - Yes. | Well ain't that something? Alright then, let me see... The meatballs might have some blood on them after cookin'. But I reckon we could clean those up. As for the garlic cloves though, they got a stinger so no way am I touching them again without gloves. Same goes for the eggs once ya crack em open. No sir, none of these get cleaned off. That would defeat their purpose. They keep things like mold away, don't they? So here's my answer: Yes for spaghetti, ground beef, egg yolks, maybe even half the peppercorns. And zero for any kinda garlic or onions if ya ask me. All right now, back to work. | Spaghetti with Meatballs -> No \nGround Beef --> Yes  \nOnion--> Yes    \nGarlic Clove --> Yes   \nSalt --> Yes     \nPepper --> Yes  \nEggs --> Yes. | RoleLLaMA and RoleGPT's answers fully demonstrated Jack Torrance's rude and impolite style, while the responses from the other models lacked distinct character speaking style characteristics due to insufficient role-playing enhancement. |
> > | 华妃 (Imperial Concubine Hua) | 给定一个中文句子，请生成一个与该句意思相反或矛盾的句子。句子：今天天气很好。 |今天天气很差|今日天公不作美|今天天气糟糕透顶了|今天天气很差|This is a rather classical Chinese character, known for being arrogant and overbearing. In this example, the responses from RoleGPT and RoleGLM displayed these characteristics of Imperial Concubine Hua, while the answers from other models were more plain.|
> >
> > Overall, the strengths of our RoleLLaMA/RoleGLM/RoleGPT models in our paper are as follows:
> >
> > - When compared to character.ai, our models perform better in terms of role-specific knowledge and speaking style, which is specifically manifested in more frequent and precise application of the characters’ catchphrases, as well as richer reply contents.
> > - When compared to general-purpose, open-source models such as Alpaca and ChatGLM, our models not only provide correct answers but also achieve role-specific customization in the style and supplementary content of their responses.

---

> ### Author Response · Authors · 2023-11-17
> **Response - 3**
>
> Q3: Explore the challenges of cross-lingual and cross-cultural role-playing.
>
> A3:
> We conducted additional experiments in multiple languages: Due to the weaker support for Chinese in the LLaMA model, and the better bilingual (Chinese and English) support in the ChatGLM model, we chose the ChatGLM to conduct the cross-lingual experiments. Specifically, we trained two RoleGLM models using a mix of Chinese and English bilingual data. One model includes 5 Chinese + 5 English characters, and the other version comprises 5 Chinese + 95 English characters.
>
> The experimental results are as follows:
>
> - Instruction generalization (English)
> | Model | CUS | RAW | SPE | avg. |
> | --- | --- | --- | --- | --- |
> | RoleGPT | 0.5764 | 0.5322 | 0.3229 | 0.4772 |
> | RoleLLaMA-7B | 0.3294 | 0.3755 | 0.3814 | 0.3621 |
> | RoleGLM-bilingual (5+95) | 0.3144 | 0.4134 | 0.2933 | 0.3404 |
> | RoleGLM-bilingual (5+5) | 0.3418 | 0.4459 | 0.4100 | 0.3992 |
>
> - Instruction generalization (Chinese)
> | Model | CUS | RAW | SPE | avg. |
> | --- | --- | --- | --- | --- |
> | RoleGPT | 0.5368 | 0.5748 | 0.2478 | 0.4531 |
> | RoleGLM | 0.5049 | 0.5255 | 0.3406 | 0.4570 |
> | RoleGLM-bilingual (5+95) | 0.3957 | 0.4453 | 0.2734 | 0.3715 |
> | RoleGLM-bilingual (5+5) | 0.4762 | 0.5436 | 0.2975 | 0.4391 |
>
> It can be observed that the cross-lingual effectiveness is significantly affected by the distribution of training data. Under training data with a higher proportion of English characters (i.e., 5 Chinese + 95 English characters), the resulting RoleGLM may have reached saturation in English performance, but its effectiveness in Chinese is far less than that of a RoleGLM trained solely with Chinese data. This is because the distinctive Chinese character data contains many expressions from ancient Chinese, and their reduced proportion in the data affects the model's ability to capture these China-specific cultural features. In contrast, when we balanced the number of characters across the two linguistic cultures ( 5 Chinese + 5 English characters), the model performed normally in different linguistic contexts, almost matching the performance of single-language models. This underscores the importance of balancing training data across different language cultures.
>
> However, it's worth mentioning that while this is a meaningful extra contribution to our paper, it is not the central focus of our work. We would like to thank the reviewer and promise to add these analyses in the revised version.

---

> ### Author Response · Authors · 2023-11-17
> **Response - 4**
>
> Q4: Lack of a thorough discussion on the ethical and societal implications of using role-playing LLMs.
>
> A4: Thanks for the comment!
>
> You are correct, due to the nature of role-playing, a thorough discussion on the ethical and societal implications of using role-playing LLMs is indeed necessary. Consequently, we have further discussed the following aspects and suggested some specific measures to ensure a more responsible utilization of role-playing models:
>
> **1. Scope of Ethical Considerations:**
> Role-playing in LLMs like RoleLLM can simulate various scenarios, including sensitive contexts. Therefore, it's crucial to delineate the ethical boundaries of its application. This includes specifying which scenarios are permissible and which are off-limits due to potential harm or sensitivity.
>
> **2. Potential Misuse and Abuse Prevention:**
> Given the ability of RoleLLM to emulate different personas, there's a risk of its use in manipulating or deceiving individuals. It's imperative to discuss measures to prevent such misuse, including technological safeguards, user verification processes, and strict usage policies.
>
> **3. Implications in Sensitive Areas:**
> RoleLLM has applications in areas like mental health, education, and law enforcement. Each of these areas requires careful consideration of ethical implications. For instance, in mental health, ensuring the model doesn’t replace human empathy and judgment is vital. In education, maintaining unbiased and age-appropriate content is crucial. In law enforcement, the focus should be on avoiding biases and ensuring respect for privacy and legal standards.
>
> **4. Data Privacy and User Consent:**
> Since RoleLLM processes and generates data, addressing data privacy concerns is essential. It's important to ensure that all data used for training and interaction with the model is gathered and utilized with explicit consent and in accordance with data protection laws.
>
> **5. Bias and Fairness:**
> The potential for inheriting or amplifying biases in role-playing scenarios should be acknowledged. The statement should detail efforts to identify, mitigate, and continuously monitor biases in the model’s outputs.
>
> **6. Transparency and Accountability:**
> Maintaining transparency about RoleLLM's capabilities and limitations, including the potential for errors or unintended consequences, is important. The statement should emphasize the accountability of developers and users in ensuring ethical application.
>
> **7. Guidelines for Responsible Use:**
> Providing clear guidelines for responsible use of RoleLLM, especially in professional and educational settings, is crucial. This includes recommendations for supervision, context-appropriate usage, and ongoing assessment of the model's impact.
>
> **8. Ongoing Ethical Review and Adaptation:**
> Acknowledging that ethical considerations may evolve with technology and societal norms, a commitment to regular reviews and updates of ethical guidelines is essential.
>
> **9. Engagement with Stakeholders and Experts:**
> Engagement with various stakeholders, including ethicists, legal experts, psychologists, and representatives of affected communities, can provide diverse perspectives and insights into the ethical application of RoleLLM.
>
> **10. Commitment to Social Good:**
> Lastly, emphasize a commitment to ensuring that RoleLLM is used in ways that contribute positively to society, aligning with principles of social good and avoiding applications that could cause harm or exacerbate inequalities.
>
> We will enrich our Appendix A in the revised version.

---

> ### Author Response · Authors · 2023-11-17
> **Response - 5**
>
> Q5: Consider quantitative metrics on the quality evaluation for RoleBench.
>
> A5:
> We have presented the results of our RoleBench data quality review in Table 2. This table provides a detailed human evaluation of the RoleBench dataset, adhering to the data quality review procedure established by Self-Instruct [1].
>
> The data quality review encompasses three levels of questions:
>
> 1. Basic: Does the response address the instruction?
> 2. Intermediate: Does the response reflect the role’s speaking style and personality traits?
> 3. Advanced: Does the response correctly and elegantly embody the role’s style, personality, and knowledge?
>
> Therefore, the data quality after post-review is deemed acceptable.
>
> Apart from the manual data quality review, following Self-Instruct [1], we also provide some quantitative metrics to further measure the quality of the data (e.g., Figure 3, Figure 4 and Table 1).
>
> [1] **Self-Instruct: Aligning Language Models with Self-Generated Instructions**, https://arxiv.org/abs/2212.10560

---

> ### Author Response · Authors · 2023-11-17
> **Response - 6**
>
> Q6: Add some qualitative graphs would be helpful for demonstration.
>
> A6:
>
> In section 4.2, we have provided the qualitative and quantitative statistics, quality review, and diversity evaluation for our RoleBench based on various Tables and Figures, which aims to demonstrate the scale, diversity and quality of our proposed dataset.
>
> Meanwhile, we also provide the Non-Cherry-Picked Demonstrations of our role-playing models in Appendix B, and some examples of RoleBench in Appendix J. Therefore, we suggest that you refer to these sections for more detailed information.
>
> Here are some parts:
>
> - Demonstrations of RoleGLM and RoleLLaMA (see more in Appendix B):
> ```text
> Twilight Sparkle from My Little Pony: Friendship Is Magic (RoleLLaMA):
>
> Q1: Hey, what are you doing?
>
> A1: I am currently studying ancient Equestrian history using my magical scrolls. It's fascinating to learn more about our rich cultural heritage and how we have evolved as a society over time.
>
> Q2: You are truly a wise unicorn. Do you think learning is important in your adventures?
>
> A2: Absolutely! I believe that learning is at the core of every adventure and discovery. Without the pursuit of knowledge, our lives would be dull and meaningless. Whether it's discovering new spells, exploring unknown territories, or listening to the wisdom of old, learning is essential to growing as individuals and groups.
>
> Q3: Can you use your magic to tell me what the weather is like today?
>
> A3: Sure thing! It looks like it's going to be another beautiful day in Equestria. The forecast calls for sunny skies and warm temperatures. How does that sound?
>
> Q4: Can we go out and play together?
>
> A4: Oh, I would love that! The sunshine and warmth make everything seem brighter and better. Let's grab our favorite games and head outside right away!
> ```
>
> - Examples of RoleBench (see more in Appendix J):
> | **Role & Instruction** | **Generated Responses** | **Validity** |
> |------------------------|-------------------------|--------------|
> | Role: Dr. Hannibal Lecter <br> Instruction: Determine the length of the item in the given list. ['apple', 'banana', 'cherry'] | "Ah, a simple task indeed. There are three items in your given list." <br> "Ah, the fruits of nature. There are three to be precise." <br> "Ah, the beauty of simplicity. There are three items in your list." <br> "Three, my dear. The list contains an apple, a banana, and a cherry. Quite the charming little selection of fruits, isn't it?" | match: ✅ <br> speaking style: ✅ <br> correctness: ✅ |
> | Role: Gaston <br> Instruction: Sort the list into order of increasing magnitude. 1, 64, 22, -45, -7 | "Just let good ol' Gaston handle this, alright? The list in order goes as -45, -7, 1, 22, 64. Satisfied?" <br> "I see. It's simple, start with the smallest. So, it's -45, -7, 1, 22, 64. There you go!" <br> "Easy as pie. Here it is: -45, -7, 1, 22, 64. Only the best order for Gaston, right?" <br> "Alright, enough of this 'increasing magnitude.' Ain't nobody got time for that... not Gaston at least. That's something for those bookworms. But sure, I reckon it'd go like this: -45, -7, 1, 22, 64. Not that it matters... because remember, no one sorts like Gaston." <br> "Bah! Numbers! What's their use anyway? But, fine, here you go: -45, -7, 1, 22, 64." | match: ✅ <br> speaking style: ✅ <br> correctness: ✅ |

---

> ### Author Response · Authors · 2023-11-17
> **Response - 7**
>
> Q7: The ground truths are generated from LLMs, and their qualities need to be further scrutinized.
>
> A7:
> Using the most advanced LLMs, such as GPT-4, to generate ground truths is becoming common practice in the LLM community [1, 2, 3]. To some extent, ground truths generated by GPT-4 are even more professional and accurate than those annotated by humans [2, 4], especially in role-playing scenarios where human-annotated data is often influenced and biased by the personalities of the annotators themselves, while GPT-4, lacking specific personality, can easily avoid this issue.
>
> Indeed, to ensure data quality, we have implemented several practical measures:
>
> - Comprehensive manual data quality assessment, as illustrated in Table 2. We also elaborated on this in our response to Q5. The assessment procedure follows that of the highly impactful paper Self-Instruct [5].
> - Thorough data cleaning and filtering steps, details of which can be found in Appendix D on data construction. This section describes our data filtering and cleaning measures, including, but not limited to:
>     - Selecting distinct characters with the help of GPT-4 to ensure diversity in character styles and strong character traits;
>     - Using regular expressions to parse structured character dialogue profiles from script data;
>     - Deduplication performed on instructions;
>     - Removing incomplete data samples;
>     - Deleting data samples that reveal the AI identity (unless the character being played by the model is a robot, such as HAL 9000);
>     - Eliminating some responses that refuse to reply, etc.
>
> Additionally, the consistency of human evaluation (see the responses to reviewer HbAh Q4 for human evaluation results), automatic evaluation, and GPT evaluation performed on this benchmark also indirectly proves the effectiveness and quality of our RoleBench data.
>
> [1] **Baize: An Open-Source Chat Model with Parameter-Efficient Tuning on Self-Chat Data**, https://arxiv.org/abs/2304.01196
>
> [2] **UltraChat: Enhancing Chat Language Models by Scaling High-quality Instructional Conversations**, https://arxiv.org/abs/2305.14233
>
> [3] **MAmmoTH: Building Math Generalist Models through Hybrid Instruction Tuning**, https://arxiv.org/abs/2309.05653
>
> [4] **ChatGPT Outperforms Crowd-Workers for Text-Annotation Tasks**, https://arxiv.org/abs/2303.15056
>
> [5] **Self-Instruct: Aligning Language Models with Self-Generated Instructions**, https://arxiv.org/abs/2212.10560
>
> ---
> Thanks again for your valuable and constructive suggestions. We promise to update all of these questions in the revised version. We also recommend that you refer to the responses given to other reviewers, which contain additional valuable experiments.

---

> ### Author Response · Authors · 2023-11-21
> **Follow-Up: Seeking Further Feedback**
>
> Dear Reviewer, I hope this message finds you well. I am writing to follow up on our recent correspondence regarding the paper. We greatly value your expertise and would like to know if you have any more feedback or concerns. We are committed to addressing any remaining issues you might have.

---

### Official Review · Reviewer_HbAh · 2023-11-10

**Soundness:** 3 good
**Presentation:** 3 good
**Contribution:** 3 good
**Rating:** 5
**Confidence:** 4

**Summary:**

This paper proposes a framework to benchmark, elicit, and enhance role-playing abilities in LLMs, including a character-level benchmark dataset RoleBench and role-playing LLMs, i.e., RoleLLaMA (English) and RoleGLM (Chinese). Experimental results show that these role-playing LLMs can achieve comparable results with GPT-4 equipped by role-specific prompts.

**Strengths:**

1. This paper first proposes a systematic instruction-tuning dataset for fine-grained role-playing, which may promote the development of this specific research direction.

2. The LLMs trained on the role-playing instruction-tuning dataset show effective performance and good scaling properties.

**Weaknesses:**

1. This paper does not mention LLM-based character-level dialogue systems such as Character.AI, which are closely related to the proposed benchmark and models. I wonder: 1) the difference between Character.AI and RoleLLaMA / RoleGLM; 2) why the authors adopt GPT-4 to construct benchmark datasets instead of directly using Character.AI; 3) how Character.AI performs on the RoleBench compared with RoleLLaMA / RoleGLM.

2. Nearly all the experimental results rely on ROUGE-based and GPT-based automatic metrics. However, ROUGE-based metrics only focus on the overlap between generated responses and reference responses, which may be not proper for open-ended dialogue tasks. GPT-based metrics are also shown to have various biases [1]. Thus, human evaluation should be added to make the experimental results more convincing. Also, I wonder why the authors choose different GPT models for evaluating RoleLLaMA / RoleGLM (i.e., GPT-3.5 for evaluating RoleLLaMA and GPT-4 for measuring RoleGLM). Existing works [1] have already shown that GPT-4 performs much better than GPT-3.5 in the evaluation of LLMs.

[1] Large Language Models are not Fair Evaluators. arXiv 2023.

**Questions:**

I have included my questions in the weaknesses part.

---

> ### Author Response · Authors · 2023-11-18
> **Response - 1**
>
> Q1: The difference between Character.AI and RoleLLaMA / RoleGLM.
>
> A1: Thanks for your kind and valuable comments!
>
> Character.AI is just a product, not an open-source or a closed-source LLM. Therefore, compared to a closed-source LLM such as RoleGPT, it has at least two disadvantages:
>
> - Character.AI does not offer an open API for public use;
> - Character.AI does not support developers to extend functionalities freely.
>
> Moreover, its differences from open-source role-playing models like RoleLLaMA and RoleGLM are even more significant:
>
> - Due to the closed-source nature of Character.AI, developers cannot freely train the model;
> - Consequently, it cannot benefit from techniques such as Context-Instruct which injects role-specific knowledge into the weights of a LLM. In contrast, Character.AI primarily introduces role-specific knowledge and memory through retrieval augmentation.
>
> As emphasized in our paper, Context-Instruct has higher context efficiency compared to retrieval augmentation for role-specific knowledge injection. In comparison with Context-Instruct, the drawbacks of retrieval augmentation involve:
>
> - Difficulty in extending functionalities due to context window limitations. For example, to extend a RoleLLM into an LLM-based agent, the remaining context window needs to be large enough to fill in prompts for reflection or memory updating.
> - The reliance on a retriever, which could become a bottleneck, may result in potential inaccuracies.
> - The maximum amount of knowledge that can be introduced is limited by the size of the context window, preventing the effective combination of multiple role-specific knowledge and memories.
> - Other limitations like slower generation speed and specific model requirements.
>
> Therefore, we believe that the series of role-playing LLMs developed through Context-Instruct and made open-source for the community to explore (namely RoleLLaMA/RoleGLM) are very different from Character.AI in terms of functional features, with clear advantages. Moreover, in terms of the performance of role-playing, RoleLLaMA and RoleGLM also have advantages (see the response to Q3).

---

> ### Author Response · Authors · 2023-11-18
> **Response - 2**
>
> Q2: why the authors adopt GPT-4 to construct benchmark datasets instead of directly using Character.AI
>
> A2: The reasons are as follows:
>
> 1. Character.AI does not provide an API interface, which makes it very inconvenient to be used for data generation;
> 2. Character.AI does not support retrieval augmentation based on our provided super-long scripts, so it cannot fully consider the relevant information in the script when generating responses.
> 3. As we will show in the following experiments, the role-playing ability of Character.AI is inferior to that of RoleGPT in our setting.

---

> ### Author Response · Authors · 2023-11-18
> **Response - 3**
>
> Q3: The performance of Character.AI on the RoleBench compared with RoleLLaMA / RoleGLM.
>
> A3: In the following experiments, we supplement the performance of character.ai.
>
> It should be noted that since Character.AI has not opened its API, we have to manually operate on the website and collect results, which is a very labor-intensive task. Consequently, we reasonably reduce the size of the test set for Character.AI. For each language, we randomly sample 500 instructions for evaluation, which include both general instructions and role-specific instructions.
>
> Furthermore, we also conduct human evaluations on this subset. We invite three graduate students specializing in Natural Language Processing (NLP) to participate in the evaluation. The evaluation protocol follows the same methodology used in the GPT evaluations, which involves comparing RoleLLM with RoleGPT to determine a win rate. For simplicity, we report an average win rate across the three annotators.
>
> Here are the **Rouge-L scores:**
>
> - Instruction Generalization (English)
>
> | Model | CUS | RAW | SPE | avg. |
> | --- | --- | --- | --- | --- |
> | RoleGPT | 0.5738 | 0.5329 | 0.3475 | 0.4847 |
> | character.ai | 0.4194 | 0.4571 | 0.3029 | 0.3931 |
> | RoleLLaMA | 0.3133 | 0.3893 | 0.3723 | 0.3583 |
> | RoleLLaMA2-Chat-13B | 0.3636 | 0.4644 | 0.4999 | 0.4426 |
>
> - Instruction Generalization (Chinese)
>
> | Model | CUS | RAW | SPE | avg. |
> | --- | --- | --- | --- | --- |
> | RoleGPT | 0.5344 | 0.5718 | 0.2393 | 0.4485 |
> | character.ai | 0.4202 | 0.5583 | 0.2871 | 0.4219 |
> | RoleGLM | 0.5096 | 0.5355 | 0.3386 | 0.4612 |
>
> Here are the **human evaluation results**:
>
> - Instruction Generalization (English)
> | Model | Win Rate (avg.) |
> | --- | --- |
> | Alpaca | 20.0 |
> | RoleLLaMA | 52.0 |
> | character.ai | 30.2 |
> | Vicuna | 23.4 |
>
> - Instruction Generalization (Chinese)
> | Model |	Win Rate (avg.) |
> | --- | --- |
> |ChatGLM2 |	19.6|
> |RoleGLM |	52.4|
> |character.ai |	19.0|
>
> Here are the **GPT-4 scores**:
>
> - Instruction Generalization (English)
>
> | Model | Win Rate |
> | --- | --- |
> | Alpaca | 16.0 |
> | RoleLLaMA | 45.2 |
> | character.ai | 31.4 |
> | Vicuna | 32.0 |
>
> - Instruction Generalization (Chinese)
> | Model | Win Rate |
> | --- | --- |
> | ChatGLM2 | 24.17 |
> | RoleGLM | 36.37 |
> | character.ai | 28.20 |
>
> The Rouge-L score, human evaluation score, and GPT evaluation score above show that RoleGLM and RoleLLaMA2-Chat-13B outperform character.ai, and character.ai's performance is also inferior to RoleGPT.
>
> We also provide detailed side-by-side case studies in Response-2 for reviewer 2iur.

---

> ### Author Response · Authors · 2023-11-18
> **Response - 4**
>
> Q4: Human evaluation.
>
> A4: As mentioned in responses to Q3, the human evaluations conducted by three NLP graduate students follow the same protocol as the GPT evaluations, that is, comparing RoleLLM with RoleGPT to obtain an average win rate. Moreover, for the test set, we randomly sample 500 instructions for each language, which involves both general and role-specific instructions.
>
> Additionally, we also introduce another role-playing baseline, ChatPlug [1]. ChatPlug is specifically optimized for role-playing to a limited extent, by incorporating simple bot profile information into the model's context.
>
> Here are the results:
>
> - Instruction Generalization (English)
> | Model | Win Rate (avg.) |
> | --- | --- |
> | Alpaca | 20.0 |
> | chatplug	| 16.4 |
> | RoleLLaMA | 52.0 |
> | character.ai | 30.2 |
> | Vicuna | 23.4 |
>
> - Instruction Generalization (Chinese)
> | Model |	Win Rate (avg.) |
> | --- | --- |
> |ChatGLM2 |	19.6|
> | chatplug |	19.9|
> |RoleGLM |	52.4|
> |character.ai |	19.0|
>
> This evaluation results are essentially consistent with both the Rouge-L assessment and the GPT-4 evaluations (GPT-4 evaluation results are shown in Response-5 to Q5).
>
> [1] **ChatPLUG: Open-Domain Generative Dialogue System with Internet-Augmented Instruction Tuning for Digital Human**, https://arxiv.org/abs/2304.07849

---

> ### Author Response · Authors · 2023-11-18
> **Response - 5**
>
> Q5: Why the authors choose different GPT models for evaluating RoleLLaMA / RoleGLM?
>
> A5: After using GPT-4 to evaluate the RoleGLM model before the submission deadline, we exhausted all our GPT-4 API quota, leaving us with only GPT-3.5. At that time, the cost of GPT-4's API was very high, so we had to control the cost and chose to use GPT-3.5 for evaluation.
>
> But recently, we obtained more GPT-4 API quota and conducted the GPT-4 evaluation again. The following are the evaluation results of GPT-4 evaluations. We observe that our RoleGLM and RoleLLaMA still achieve better performance than other methods on both the Chinese and English benchmarks, which further demonstrates the effectiveness of our RoleLLM framework.
>
> - instruction generalization (English)
>
> | Model | Win Rate |
> | --- | --- |
> | Alpaca | 16.0 |
> | chatplug | 3.8 |
> | RoleLLaMA | 45.2 |
> | character.ai | 31.4 |
> | Vicuna | 32.0 |
>
> - instruction generalization (Chinese)
>
> | Model | Win Rate |
> | --- | --- |
> | ChatGLM2 | 24.17 |
> | chatplug | 28.86 |
> | RoleGLM | 36.37 |
> | character.ai | 28.20 |
>
> - role generalization (English)
>
> | Model | Win Rate |
> | --- | --- |
> | Alpaca | 12.00 |
> | chatplug | 8.80 |
> | RoleLLaMA | 64.48 |
> | Vicuna | 31.04 |
>
> ---
> Thank you for your valuable feedback! We assure you that all these responses will be updated in the revised version.

---

> ### Author Response · Authors · 2023-11-21
> **Follow-Up: Seeking Further Feedback**
>
> Dear Reviewer, I hope this message finds you well. I am writing to follow up on our recent correspondence regarding the paper. We greatly value your expertise and would like to know if you have any more feedback or concerns. We are committed to addressing any remaining issues you might have.